# Reversible intercalation of methyl viologen as a dicationic charge carrier in aqueous batteries

Zhixuan Wei[1,2], Woochul Shin[2], Heng Jiang[2], Xianyong Wu[2], William F. Stickle[3], Gang Chen[1], Jun Lu[4], P. Alex Greaney[5], Fei Du[1] & Xiulei Ji[2]

The interactions between charge carriers and electrode structures represent one of the most important considerations in the search for new energy storage devices. Currently, ionic bonding dominates the battery chemistry. Here we report the reversible insertion of a large molecular dication, methyl viologen, into the crystal structure of an aromatic solid electrode, 3,4,9,10-perylenetetracarboxylic dianhydride. This is the largest insertion charge carrier when non-solvated ever reported for batteries; surprisingly, the kinetic properties of the (de) insertion of methyl viologen are excellent with 60% of capacity retained when the current rate is increased from 100 mA g$^{-1}$ to 2000 mA g$^{-1}$. Characterization reveals that the insertion of methyl viologen causes phase transformation of the organic host, and embodies guest-host chemical bonding. First-principles density functional theory calculations suggest strong guest-host interaction beyond the pure ionic bonding, where a large extent of covalency may exist. This study extends the boundary of battery chemistry to large molecular ions as charge carriers and also highlights the electrochemical assembly of a supramolecular system.

[1] Key Laboratory of Physics and Technology for Advanced Batteries (Ministry of Education), State Key Laboratory of Superhard Materials, College of Physics, Jilin University, 130012 Changchun, People's Republic of China. [2] Department of Chemistry, Oregon State University, Corvallis, OR 97331, USA. [3] Hewlett-Packard Co, 1000 NE Circle Blvd., Corvallis, OR 97330, USA. [4] Chemical Sciences and Engineering Division, Argonne National Laboratory, Argonne, IL 60439, USA. [5] Materials Science and Engineering, University of California, Riverside, CA 92521, USA. Correspondence and requests for materials should be addressed to J.L. (email: junlu@anl.gov) or to P.A.G. (email: agreaney@engr.ucr.edu) or to F.D. (email: dufei@jlu.edu.cn) or to X.J. (email: david.ji@oregonstate.edu)

To store intermittent solar and wind energy efficiently calls on exploration of new battery chemistries that go beyond the conventional domains of solid-state ionics based on topotactic reactions with metal ions as charge carriers. Indeed, it is the ion charge carrier that often dictates the nature of the battery chemistries. Among ion charge carriers, metal ions have come in succession from alkali metal ions[1–11] to alkaline earth metal ions[12,13] to Al[3+][14], and Zn[2+][15,16], being explored for next generation batteries. However, the battery community has given limited attention to non-metal cations. Recently, small non-metal ions such as proton, hydronium[17], and ammonium[18] have been demonstrated as competitive charge carriers. Along this line, there exist possibilities to explore new charge carriers for energy storage systems, particularly the non-metal molecular ones.

To date, most battery chemistries are considered solely associated with the ionic bonding between the metal-ion charge carriers and electrode hosts even though there might be some covalent contribution, which would play so minor a role. Nevertheless, the assumption of pure ionics may not be held true if large non-metal charge carriers, aromatic molecules in particular, are employed. One of the intermolecular interactions beyond pure ionicity is the $\pi$–$\pi$ interaction, where it is intriguing to couple a pair of ion and electrode to play out this interaction during the charge/discharge processes of rechargeable batteries.

Thread-like molecular dications, e.g., $1,1'$-dialkyl-$4,4'$-bipyridinium, which are known as "viologens", are a class of $4,4'$-bipyridyl derivatives. The unique $\pi$-framework of the bipyridyl nucleus makes it possible to interact with electrode structures via their aromatic units. On the other hand, aromatic molecular solids have caught attention as reversible electrode materials for various types of battery chemistries[19–24]. Organic crystals are assembled by $\pi$–$\pi$ aromatic stacking, which, compared to the inorganic materials, typically possess larger interlayer spacings, and more flexible solid structures. Among aromatic solid electrode materials, 3,4,9,10-perylenetetracarboxylic dianhydride (PTCDA), a well-known red pigment, has been investigated as the electrode host for various ions, including metal ions of Li$^+$, Na$^+$, K$^+$, Mg$^{2+}$, and non-metal hydronium and ammonium[17,25–27]. PTCDA has a perylene core and one anhydride group on each of the both ends of perylene. The aromatic structures of viologen and PTCDA provide an interesting ion/electrode combination to study the possible $\pi$–$\pi$ interaction chemistry between the charge carriers and electrodes (Fig. 1).

Herein, we report the reversible topotactic (de)intercalation chemistry of a molecular ion, methyl viologen, $(1,1'$-dimethyl-$4,4'$-bipyridinium, hereafter denoted as MV$^{2+}$) inside a highly crystalline PTCDA electrode. This reaction exhibits a reversible specific capacity of 105 mAh g$^{-1}$ and a fairly stable cycle life. Both ex situ X-ray diffraction (XRD) and ex situ Fourier transform infrared spectroscopy (FTIR) studies provide strong evidence of the host-guest chemistry of methyl viologen inside PTCDA.

**Fig. 1** Molecular structures of the charge carrier and the host material. **a** MV$^{2+}$; **b** PTCDA

## Results

**Electrochemical performance and determination of intercalant.** The crystal structure of PTCDA was studied by powder XRD. As shown in Supplementary Fig. 1, PTCDA exhibits a highly crystalline structure, which can be indexed to monoclinic P2$_1$/c space group. Scanning electron microscopy (SEM) reveals one-dimension rod-like morphology of PTCDA (Supplementary Fig. 2). The rods are ~1 μm in length and ~200 nm in width. We first investigated the (de)insertion of MV$^{2+}$ in the PTCDA electrode by galvanostatic charge/discharge (GCD) cycling in a three-electrode cell setup, which comprises an excessive mass of activated carbon as the counter electrode, an Ag/AgCl reference electrode in 3 M KCl aq. (0.21 V vs. SHE), and 0.1 M aqueous solution of methyl viologen dichloride (pH ≈ 3.5) as the electrolyte. The oxidation and reduction of the working electrode (PTCDA) is hereafter denoted as the charge and discharge process, respectively. The lower cutoff potential is chosen to prevent a severe hydrogen evolution reaction (HER) as well as the MV$^{2+}$ plating on the electrode surface (Supplementary Fig. 3). As Fig. 2a shows, PTCDA exhibits the charge and discharge capacity of 90.0 mAh g$^{-1}$ and 125.5 mAh g$^{-1}$, respectively, in the first cycle. It displays generally sloping discharge profiles but two sequential plateaus in the charge processes. Note that the discharge profile in the first cycle shows higher polarization than in the following cycles. This may come from the more significant "effort" of PTCDA crystals to accommodate MV$^{2+}$, as suggested by the significantly altered morphology after the 1st cycle (Supplementary Fig. 4). Importantly, such a conditioning process is desirable since the average operation potential of MV$^{2+}$ intercalation in the following cycles is enhanced. Besides, the first discharge shows an irreversible plateau at −0.3 V, which is due to the irreversible reduction of the impurities on the carbon fiber paper as the current collector (Supplementary Fig. 5). This is largely responsible for the low first-cycle Coulombic efficiency (CE) of 71%. From the 2nd cycle, the CE rises to 93% and becomes stable afterwards, while the specific charge capacity slightly increases compared to the first cycle. After 30 cycles, a discharge capacity is stable at 105.7 mAh$^{-1}$, as shown in Fig. 2b. We tentatively attribute the reversible capacity to the insertion of MV$^{2+}$ into the structure of PTCDA. As one piece of supporting evidence, in one controlling experiment, as shown in Supplementary Fig. 6, when graphite serves as the electrode, there is barely any reversible capacity exhibited, where the CE in the first cycle is <20%.

As the electrolyte is mildly acidic, it is likely that PTCDA reversibly stores hydronium as well in addition to MV$^{2+}$. To investigate this aspect, we tested GCD of PTCDA in a diluted H$_2$SO$_4$ electrolyte with pH of 3.5 the same as the electrolyte of MV$^{2+}$, which delivers a reversible capacity of 12 mAh g$^{-1}$, 11.4% of the whole capacity (Supplementary Fig. 7). Thus, we estimate the capacity of pure reversible MV$^{2+}$ storage as 78 mAh g$^{-1}$ in the first cycle, which suggests that 0.57 MV$^{2+}$ is reversibly incorporated per PTCDA molecule (based on the theoretical capacity of 137 mAh g$^{-1}$ if a two-electron transfer occurs per PTCDA molecule). Meanwhile, a similar CE (92.9%) in this case indicates that HER on PTCDA that can hardly be fully eliminated in this mildly acidic electrolyte is responsible for the low CE in MV$^{2+}$-based electrolyte.

To confirm the intercalant, we conducted in situ electrochemical quartz crystal microbalance (EQCM) measurements recorded during a typical cathodic scan of CV tests. As shown in Fig. 2c, the mass evolution during the discharge process proceeds in two stages. During the first stage, the electrode mass increases at a rate of 47.5 g/mol e$^-$, indicative of co-intercalation of MV$^{2+}$ and hydronium, H$_3$O$^+$. As for the second stage from approximately −0.53 V to the potential of fully discharged (Supplementary Fig. 8), the electrode has a mass gain at 92.3 g/mol e$^-$.

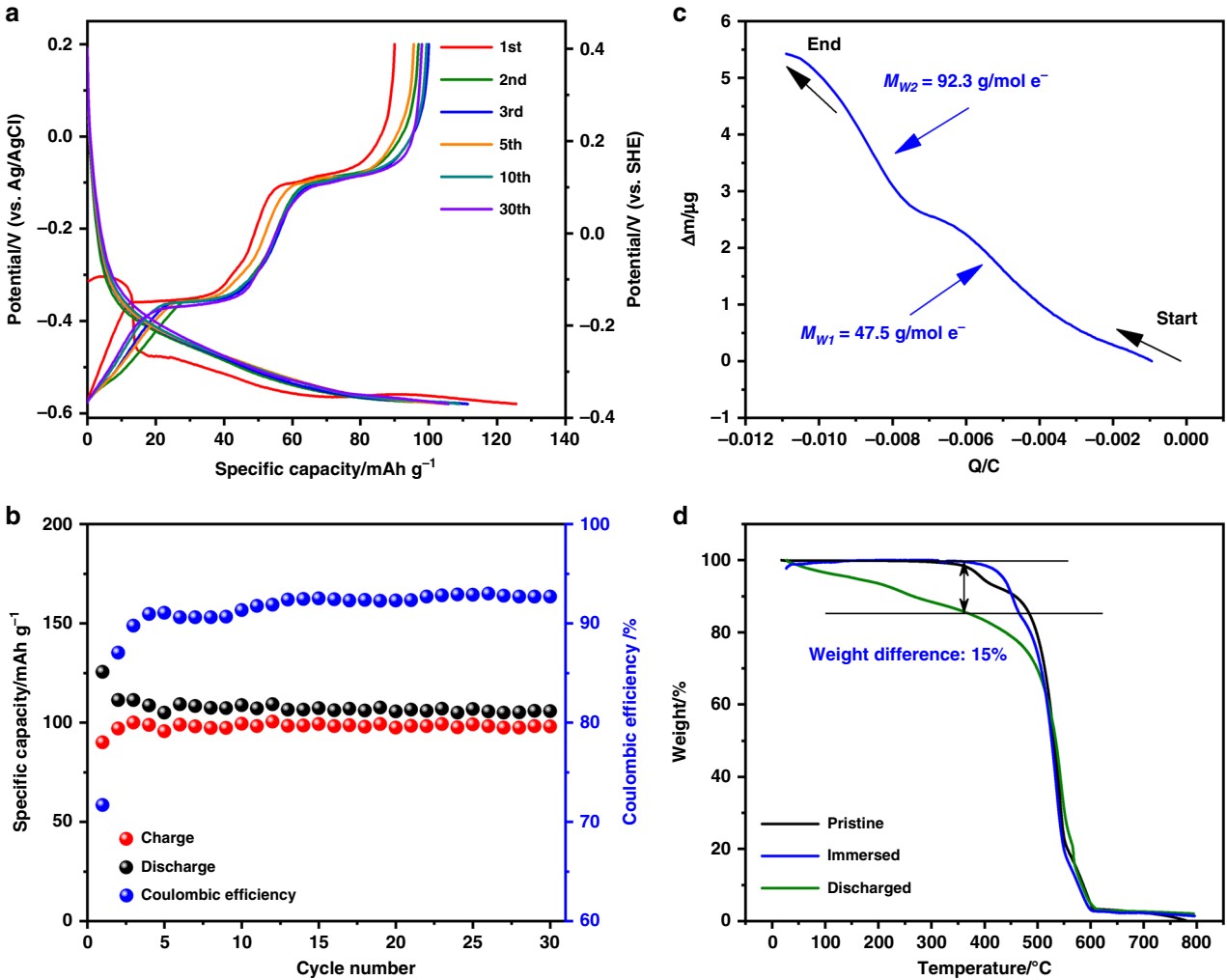

**Fig. 2** Electrochemical performance of MV$^{2+}$ intercalation in PTCDA electrodes. **a** GCD potential profiles of PTCDA between −0.58 and 0.2 V (vs. Ag/AgCl) at a current rate of 100 mA g$^{-1}$; **b** Galvanostatic cycling of the PTCDA electrode at 100 mA g$^{-1}$ for 30 cycles; **c** EQCM curve recorded during a typical cathodic CV scan; **d** The TGA profiles of the pristine electrode, the electrolyte-immersed electrode, and the discharged electrode tested under air flow

Considering the molar mass of MV$^{2+}$ of 186 g mol$^{-1}$ and its divalency, this unequivocally demonstrates the intercalation of one MV$^{2+}$. Furthermore, we also carried out thermogravimetric analysis (TGA) of the electrode after it was fully discharged to verify the inserted charge carrier. As shown in Fig. 2d, we determined the additional mass added during the discharge process to be 15 wt.%, compared to the pristine PTCDA. This corresponds to a ratio of 1:0.53 between the mass of PTCDA and that of MV$^{2+}$, corroborating the mass of the reversibly inserted MV$^{2+}$ estimated by the GCD results. This is significantly heavier than the case if the charge carrier is solely proton.

One concern is whether the mass gained during discharge is due to the ion adsorption to the surface of the PTCDA electrode. As a control experiment, we immersed the PTCDA electrode in the MVCl$_2$ electrolyte for 3 days, and performed another TGA test afterwards. As illustrated by the blue profile in Fig. 2d, there is nearly no additional weight loss in TGA compared to the non-soaked electrode. Besides, the Brunauer–Emmett–Teller (BET) surface area of PTCDA is 13.29 m$^2$ g$^{-1}$ (Supplementary Fig. 9). Since the electrochemical surface adsorption positively correlates with the specific surface area, it would not make significant contribution in this case. Additionally, to further confirm the TGA results, CHN elemental analysis was also conducted. As

shown in Supplementary Table 1, the results demonstrate that 0.579 MV$^{2+}$ is incorporated per PTCDA molecule, which is in well accordance with the GCD results.

Surprisingly, the reversible MV$^{2+}$ intercalation exhibits a higher operation potential and a much smaller extent of polarization compared with several metal and non-metal ions such as sodium-, potassium-, magnesium-, and ammonium-ions (Supplementary Fig. 10), suggesting superior thermodynamic and kinetic feasibility for PTCDA crystals to host MV$^{2+}$ ions. The higher operation potential mitigates HER in acidic electrolytes.

**Interaction between MV$^{2+}$ and the MV$^{2+}$-intercalated PTCDA.** MV$^{2+}$ insertion causes reversible structural changes of the PTCDA crystals. As shown by the ex situ XRD patterns taken at different state of charge (SOC) in the first cycle (Fig. 3b), upon discharge to −0.55 V (Stage 2), new minor peaks appear at 10.7° and 29.4°, indicating the formation of a new phase. With further MV$^{2+}$ insertion (Stage 3), the new peak at 10.8° becomes more intense while the (021) peak gets weakened, strongly suggesting that a new phase of the MV-PTCDA intercalation compound emerges, the new peak can be assigned to (02$\bar{1}$) plane, which will be discussed later. Furthermore, the pristine PTCDA structure is

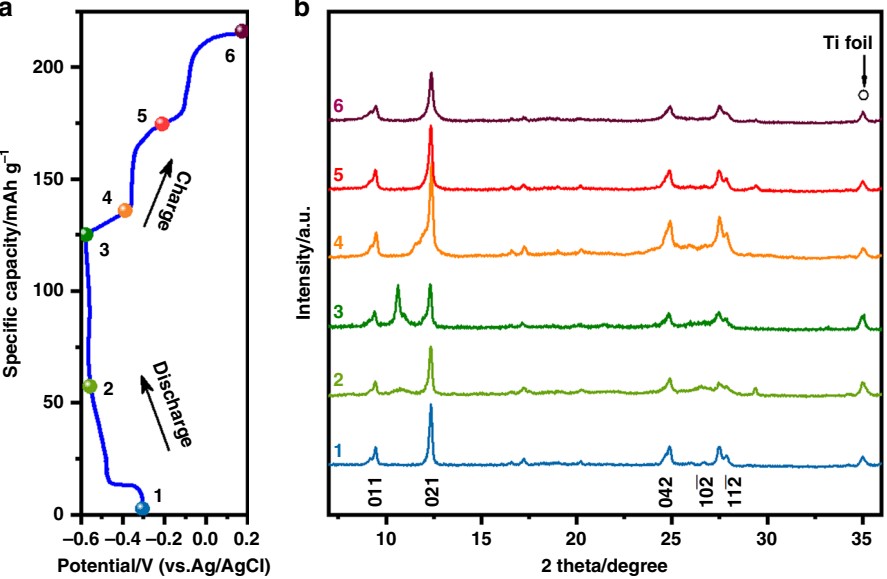

**Fig. 3** Structural evolution of PTCDA during $MV^{2+}$ intercalation. **a** GCD potential profiles of the PTCDA electrode in the first cycle; **b** Ex situ XRD patterns of the PTCDA electrode recorded at different SOC

well reserved since the (011), (042), (10$\bar{2}$), and (11$\bar{2}$) peaks of PTCDA remain unchanged. Besides, the peaks in the XRD patterns of $MVCl_2$ and the fully reduced $MV^{2+}$ ($MV^0$) are different from those of the MV-PTCDA compound, as shown in Supplementary Fig. 11, again indicating the formation of a new compound.

During the following charge, all the newly formed peaks gradually mitigate and vanish, and the intensity of the original peaks are recovered. Along this line, ex situ transmission electron microscopy (TEM) and selected area electron diffraction (SAED) studies are performed to gain insights into the microstructural transformation (Supplementary Fig. 12). At the end of discharge, the diffraction spot of the (02$\bar{1}$) peak emerges, which disappears during the following charge process. The results again confirm the reversible formation of the new phase.

Moreover, the evolution of the XRD peaks of the PTCDA electrode at the 5th cycle and the 20th cycle is almost identical as the first cycle except for the peak intensity, as shown in Supplementary Fig. 13, suggesting the surprising reversibility of structural changes of the PTCDA crystals upon repetitive hosting $MV^{2+}$. It is worth noting that the discharged samples in the subsequent cycles all feature amorphization, which reflects the relatively weak interactions, i.e., van der Waals forces, among the PTCDA molecules that form the crystalline structure, as often observed for organic electrodes[28–30].

It is intriguing that the discharge profiles are slopes but the following charge processes depict plateau profiles. From the XRD patterns, as shown in Supplementary Fig. 13, the discharge process transforms the structure completely amorphous, which, however, returns back to a crystalline structure during the following charge process. The nuances are that the amorphous-to-crystalline structural change coincides with plateau profiles, whereas the crystalline-to-amorphous transition coexists with the sloping profiles[31,32]. Such interesting structural changes reveal that the molecular structure of PCTDA is not only flexible to accommodate the incorporation of large ions but also elastic upon the removal of these ions.

Ex situ Fourier transform infrared spectroscopy (FTIR) measurements reveal the impacts of hosting the guest $MV^{2+}$ inside PTCDA electrodes after discharge and the following charge. As shown in Fig. 4a, the peaks between 1726–1766 cm$^{-1}$

ascribed to carbonyl groups in PTCDA decrease in intensity upon discharge, indicative of the weakened C=O bonds on the reduced PTCDA molecules. Furthermore, different from the insertion of $Na^+$, $K^+$, and $Mg^{2+}$ into PTCDA that causes enolation and the associated new peak formation near 1800 cm$^{-1}$[27], it can be conjectured that the binding of large $MV^{2+}$ to PTCDA molecules may not be localized on the sites of C=O bonds; but being more delocalized (Fig. 4c). To better understand the electron transfer, we performed ex situ X-ray photoelectron spectroscopy (XPS) on O 1s to monitor the bonding evolution upon discharge and charge processes, which turns out to be highly reversible (Fig. 4b). The spectra exhibit two peaks at 534 eV and 531.5 eV, denoting C-O and C=O, respectively. Both peaks red-shift in the discharged state, reflecting the increase in electron density on oxygen by accepting electrons. Besides, the bonding of $MV^{2+}$ in the PTCDA structure is suggested by the new vibration peak at 1454 cm$^{-1}$ for the stretching mode of the C–N bond from $MV^{2+}$[33]. In addition, the new peaks at 1633 cm$^{-1}$, 1557 cm$^{-1}$, 1424 cm$^{-1}$, 1377 cm$^{-1}$, and 1347 cm$^{-1}$ (marked as red triangles) that belong to neither the pristine PTCDA nor the MV-dichloride solid nor $MV^0$ tell new chemical interactions formed inside the MV-PTCDA intercalation compound. Among them, the peaks at 1424 cm$^{-1}$ and 1347 cm$^{-1}$ can be assigned to the stretching vibration of C-N[34,35]; the peak at 1377 cm$^{-1}$ is related to the deformation vibration of the terminal $CH_3$ groups[36,37], while the band at 1557 cm$^{-1}$ is associated with the C–H stretching of 4-substituted pyridine, and the band at 1633 cm$^{-1}$ is attributed to a combination of C–H and C–N stretching[38]. After the following charge process, the original FTIR spectrum of PTCDA recovered and all the newly formed peaks disappeared, which again suggests that the $MV^{2+}$ insertion is highly reversible. Combining the above discussion and the electrochemical results, the postulated redox mechanism is illuminated in Fig. 4c.

**First-principles calculations of the host-guest system.** The collective electrochemical, TGA, XRD, and FTIR results provide compelling evidence that $MV^{2+}$ ions are reversibly inserted into the structure of the PTCDA electrode. However, two more important questions beg answers: on what sites the large $MV^{2+}$

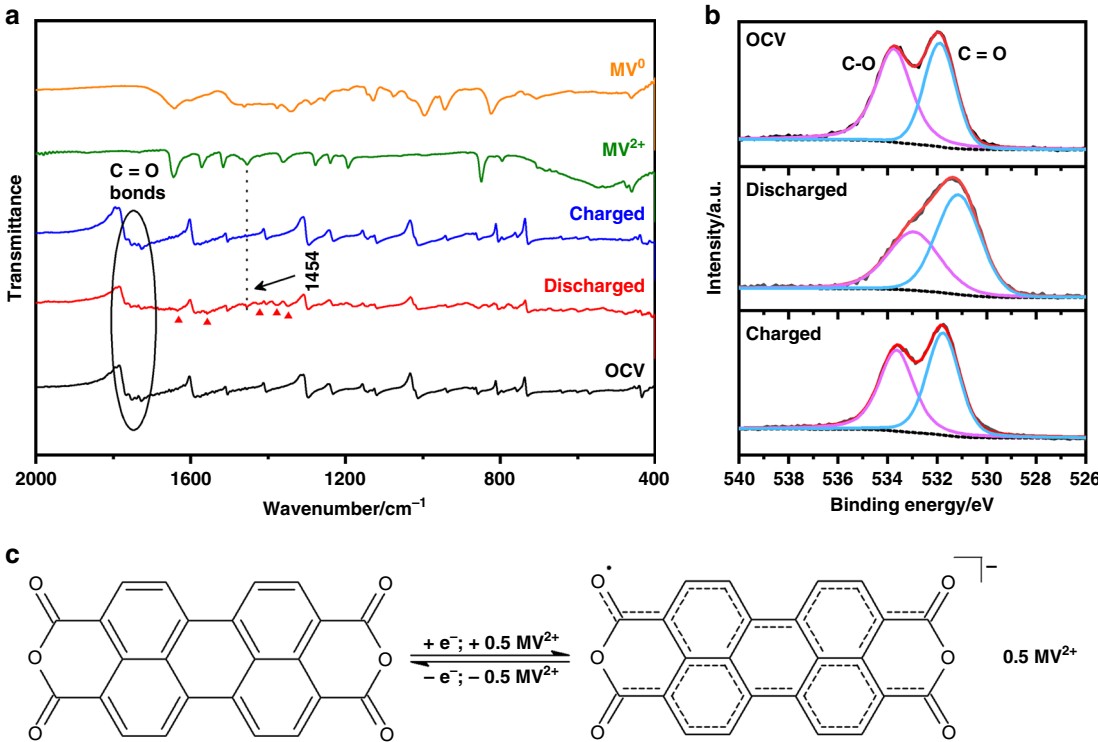

**Fig. 4** Analysis of the reaction mechanism during the electrochemical process. **a** FTIR spectra of the PTCDA electrode recorded at different SOC: OCV (black), the initial discharged (red), the charged state (blue), and FTIR spectra of methyl viologen dichloride powder (green) and its reduced form of MV$^0$ (orange); **b** XPS O 1s spectrum of PTCDA electrodes recorded at different SOC; **c** Schematic illustration showing the possible reversible electrochemical redox mechanism of PTCDA electrode

ions are located in the PTCDA crystals and what nature the chemical interaction is between the guest and the host. To shed light on these critical issues, we performed first-principles density functional theory (DFT) calculations on the simulated MV$^{2+}$-PTCDA systems. Since the MV$^{2+}$ cation is much bigger than a typical metal ion, there are mainly two possible insertion sites: (1) sandwiched between PTCDA molecules and (2) normal to the PTCDA planes between the columns of stacked PTCDA in a herringbone structure. Thus, the MV$^{2+}$ molecules were placed in several plausible insertion sites, and then we let the systems to be fully relaxed to find an energetically favorable configuration, where we find out which structures match with the experimental XRD results better.

According to the molar ratio between guest MV$^{2+}$ and the PTCDA host from the TGA results, we computed the following stoichiometry: one PTCDA unit cell hosts one MV$^{2+}$, where one unit cell contains two PTCDA molecules. First, we put MV$^{2+}$ cations in parallel to the PTCDA molecules, followed by structural relaxation until the force on all atoms reduces to <0.02 eV Å$^{-1}$. Figure 5 displays geometries of sandwiched structures after energy minimization and the corresponding XRD patterns. Since the interplanar distance in pristine PTCDA is not large enough to host MV$^{2+}$ ions, PTCDA molecules as well as MV$^{2+}$ cations are forced to deform, as indicated in Fig. 5a, which is energetically unfavorable although the generated XRD peaks are in the proximity of the experimental one. To alleviate the disadvantage, we manually expanded the PTCDA interplanar space and relaxed the structure again (Fig. 5b). It still distorts the PTCDA domains, and the generated XRD patterns poorly match with the experimental one.

As for the second types of sites, we inserted MV$^{2+}$ into the crevices between the columns of the stacked PTCDA molecules. Here, the angles between MV$^{2+}$ and the PTCDA column vary

from 0° to 45° to 90° to explore multiple insertion conditions, as shown in Fig. 6. All the MV$^{2+}$ ions are aligned near the oxygen terminals of PTCDA molecules, owing to its hydrogen bonding with MV$^{2+}$. After structural relaxation, the generated XRD patterns have been compared to the experimental data.

As shown in Fig. 7, at the parallel-to-column position or at the normal-to-column position, the simulated XRD patterns fit the experimental results poorly. However, when we tilted MV$^{2+}$ to be 45°, and surprisingly, the generated XRD pattern matches with the discharged-state XRD pattern very well (the stage 3 in Fig. 3). Peaks split into (011), (02$\bar{1}$), and (021), and they have similar relative intensity with the experimental output (Fig. 7c). There is some angle shift, which we postulate is due to the fact that the simulation is done 'under vacuum conditions', where the lattice $d$-spacings are different from the samples under ambient conditions. Generally, this tilted model can provide some valuable insights on how the MV$^{2+}$ ions are stored and oriented with respect to PTCDA's structure.

To validate this model, we extracted one MV$^{2+}$ out of the unit cell to reach the stoichiometry of 3PTCDA:1MV$^{2+}$ and simulated the corresponding XRD pattern (Supplementary Fig. 14). The simulated XRD peaks match well with stage 4 and stage 5 (the half-charged stage in Fig. 3), which represents the de-insertion process. Therefore, MV$^{2+}$ ions are most likely inserted at 45° into the crevices between the columns of the stacked PTCDA molecules.

In addition, we conducted Bader charge analysis on MV$^{2+}$ to understand the interactions between the host and the inserted species, where in the 45° MV$^{2+}$-inserted model, the oxidation state of MV is +1.31 instead of +2, where nearly 0.7 electrons are transferred from PTCDA to one MV$^{2+}$, and this may explain the driving force of vertical insertion. We also did the Bader charge analysis on the PTCDA host, which would be −1

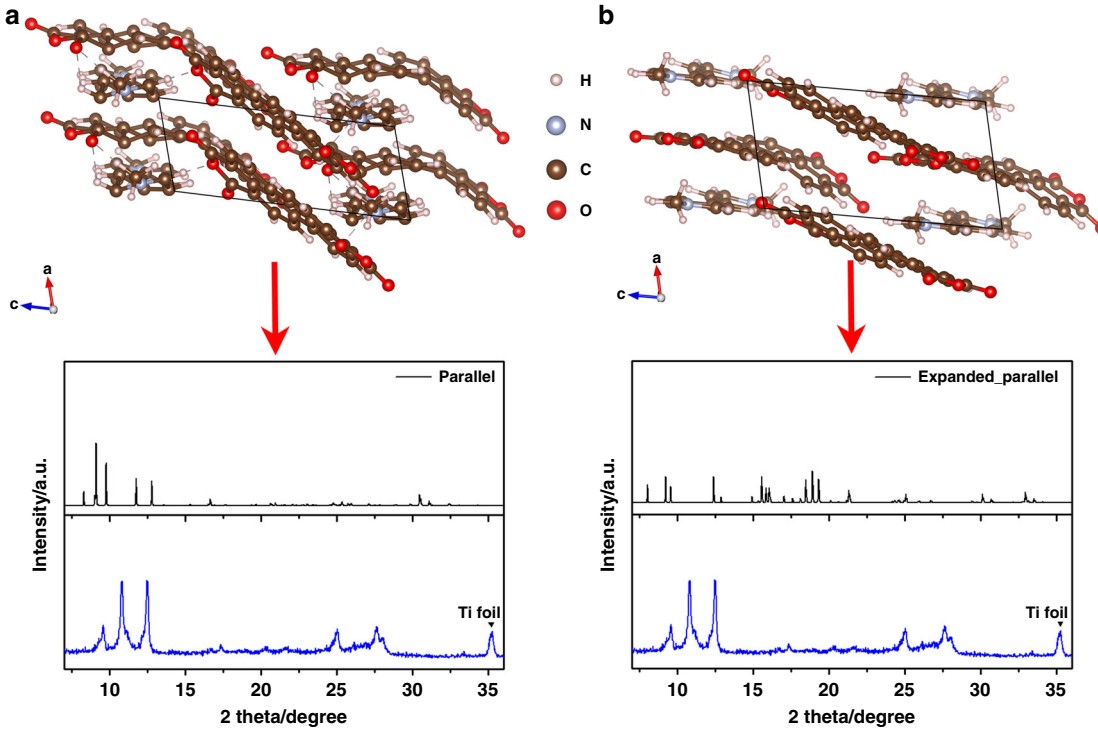

**Fig. 5** Simulated $MV^{2+}$ inserted PTCDA in sandwiched geometries and the corresponding XRD patterns after energy minimization: **a** Parallel insertion into the pristine PTCDA domains; **b** Parallel insertion after PTCDA interplanar distance extension along *a*-axis

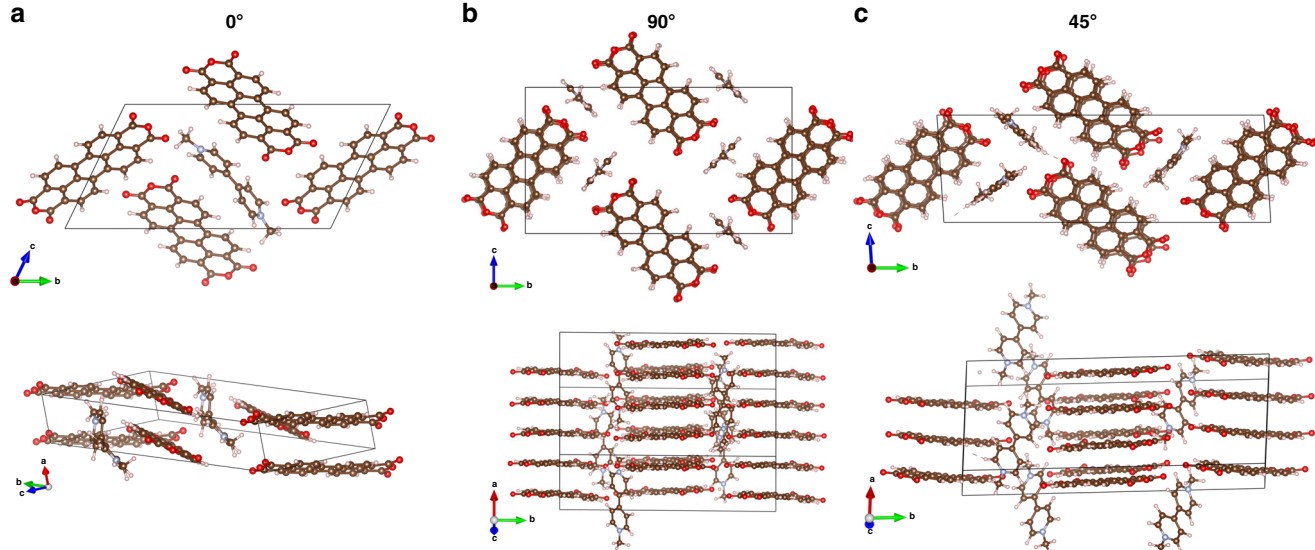

**Fig. 6** Models of simulated $MV^{2+}$-inserted PTCDA in vertical geometries from top view (top) and side view (bottom): **a** 0°; **b** 90°; **c** 45°. Angles are indicating relative slope to PTCDA domains

according to the ratio of 1/2 between $MV^{2+}$ and PTCDA if there were no charge transfer. The calculated Bader charge of PTCDA turns out to be −0.69. Thus, 0.31 electrons are transferred from each of the two PTCDA molecules to $MV^{2+}$ and the Bader charge of $MV^{2+}$ is 1.38, corroborating the above calculation. It seems that $MV^{2+}$ should be reduced first to $MV^{1.31+}$ or $MV^{1.38+}$, and then PTCDA is reduced afterward from the above Bader charge calculations. However, as a single discrete molecule of $MV^{2+}$, it cannot possibly take in $0.69e^-$ from the surface of PTCDA electrode because electrons are indivisible in electrochemical reactions. Therefore, it is more likely that during the $MV^{2+}$ insertion, electrons are received by PTCDA molecules first, and

the PTCDA molecules would pass these electrons to the intercalated $MV^{2+}$ via the charge transfer, instead of the direct reduction of $MV^{2+}$ itself on the surface of PTCDA.

We also conducted ex situ XPS analysis on nitrogen to monitor the electron transfer in the complex. As shown in Supplementary Fig. 15, the valence state of nitrogen indeed got decreased after intercalating into the PTCDA structure.

**Reaction kinetics studies**. To further investigate the kinetic properties of storing $MV^{2+}$ inside PTCDA crystals, we first tested the rate capability of the $MV^{2+}$-PTCDA electrode. As shown in

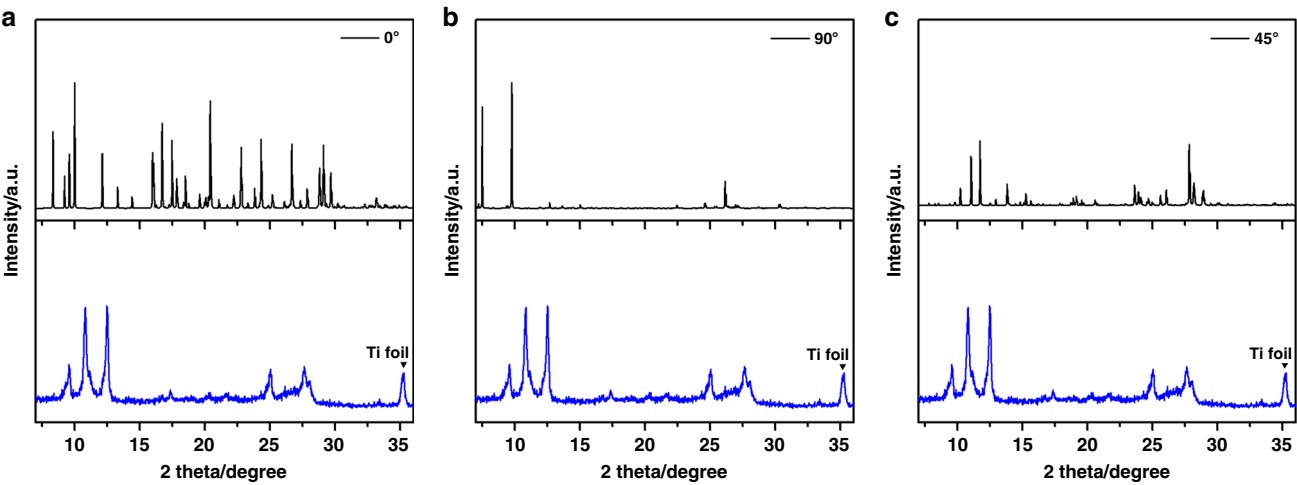

**Fig. 7** Simulated XRD patterns of vertically inserted models: **a** 0°; **b** 90°; **c** 45°. Angles are indicating relative slope to PTCDA domains

**Fig. 8** Rate performance and the reaction kinetics. **a** Rate performance and CE of $MV^{2+}$ storage in PTCDA from 100 mA g$^{-1}$ to 2000 mA g$^{-1}$; **b** The GCD potential profiles of the PTCDA electrode tested at different current rates; **c** The GITT potential profile of the PTCDA electrode in the 31st charge process; **d** Diffusion coefficients calculated from the GITT potential profiles as a function of SOC during the 31st charge

Fig. 8a, at a high current rate of 2000 mA g$^{-1}$, a specific capacity of 66.7 mAh g$^{-1}$ can still be retained, and the shape of the GCD potential profiles can be well retained upon increasing the current rate (Fig. 8b). These features suggest fast transport of $MV^{2+}$ in the cells despite $MV^{2+}$'s large size. Note that the disparity in the CE under different current rates can be attributed to the extents of the HER on the PTCDA electrode. As the current rate

increases, the onset potential of HER is lowered due to polarization, thus inhibiting HER from taking place and raising the CE to nearly 100%. Interestingly, the rate performance of storing $MV^{2+}$ in PTCDA is comparable with that of hosting metal-ions and ammonium-ions, as shown in Supplementary Figs. 16 and 17, which again indicates the good kinetics of $MV^{2+}$ (de) intercalation.

Galvanostatic intermittent titration technique (GITT) measurements were carried out on the 31st charge process to gain insight into the kinetics of $MV^{2+}$ diffusion in PTCDA. Figure 8c, d show the overall trend of the diffusion coefficient, which is found strongly dependent on the specific SOC, and the minimal diffusivity values correspond to the potential plateaus of the charge process. Surprisingly, despite the large size of the $MV^{2+}$ molecule, the diffusion coefficient is comparable with ions in aqueous ion batteries, such as $Zn^{2+}$ as a charge carrier[39,40]. Such an excellent diffusion behavior can be ascribed to the low dehydration energy penalty at the electrode/electrolyte interface. Specifically, in aqueous electrolytes, the water molecules that surround large ions, like $MV^{2+}$, are largely hydrogen bonded and are more mobile than the case of small metal ions[41]. Hence, this facilitates the ion transport through the electrode/electrolyte interface. Furthermore, the cycle stability of hosting $MV^{2+}$ was also evaluated in comparison with the performance of other ions. As displayed in Supplementary Fig. 18, after 100 charge/discharge cycles at 200 mA g$^{-1}$, a charge capacity of 78 mA h g$^{-1}$ is retained, corresponding to capacity retention of 86%. Of note, the storage of $MV^{2+}$ in PTCDA demonstrates much greater stability comparing with other ions, as illustrated in Supplementary Fig. 19.

## Discussion

A question is why the discharge potential for PTCDA to host $MV^{2+}$ is higher than hosting other metal ions. There are two major factors that determine the potential of ion insertion: (1) the desolvation energy of the ion charge carrier from the aqueous electrolyte and (2) the binding energy between the inserted ions and the host structure. The desolvation energy of the much larger $MV^{2+}$ is understandably lower than the smaller metal ions, where by compensating a smaller energy penalty for desolvation, the insertion potential is raised. Note that it can be very interesting to theoretically understand how the desovlation of large molecular ions such as $MV^{2+}$ affects the electrochemical properties of host electrodes in comparison with smaller metal ions. However, considering the scope of this work and the primary theme about what takes place after $MV^{2+}$ ions migrate into the PTCDA structures, such theoretical studies can be conducted in the future.

On the other hand, $MV^{2+}$ is redox active, where based on Bader charge analysis on the inserted $MV^{2+}$ (45° tilted), the oxidation state of MV is +1.31 or +1.38 instead of +2, where nearly 0.7 electrons are transferred from PTCDA to one $MV^{2+}$. Such an electron transfer from the host to the inserted ion suggests a high extent of the polar covalent bonding between the MV charge carrier and the PTCDA host. The strong guest–host bonding would lower the total energy of the electrode system, thus raising the insertion potential. Therefore, these two factors may explain why the discharge potential of $MV^{2+}$ is higher than the insertion of smaller metal ions.

Another interesting aspect is that the storage of much larger $MV^{2+}$ in PTCDA exhibits more stable cycling performance and slightly better rate capability compared to other smaller metal ions (Supplementary Figs. 16 and 17). This phenomenon points to the solvated ion size, which we postulate is responsible for our observation. Generally, due to the high-charge density, small metal ions are surrounded by water molecules to form bulky and clumsy solvation spheres, giving rise to large desolvation energy penalty at the electrode/electrolyte interfaces during ion insertion. For instance, the smallest metal ion, lithium ion, suffers from large overpotentials in the aqueous electrolytes due to the large size of the hydrated lithium ion, which cannot even survive the second cycle of the (de)insertion into the PTCDA electrode (see Supplementary Fig. 20). On the other hand, $MV^{2+}$ are vertically inserted into the slits of the herringbone structure of PTCDA, where the effective ionic size is essentially the thickness of the planar molecule, being much smaller than the hydrated spheres of metal ions. This may facilitate the ionic transportation in the host crystals, and thus promote the better rate capability and cycling stability.

As the last note, an intriguing question is whether molecules larger than $MV^{2+}$ can be employed as the ion charge carriers for the PTCDA electrode. To shed some light on this interesting question, we tested the electrochemical (de)insertion of ethyl viologen ($EV^{2+}$) into the same PTCDA electrode. Remarkably, the reversible capacity is around 125 mAh g$^{-1}$, which is even slightly higher than $MV^{2+}$ (Supplementary Fig. 21). Thus, we stay optimistic that even larger molecules may be inserted into PTCDA, where such an electrochemical reaction is, in fact, a powerful synthesis method of assemblies of supramolecular solids.

In conclusion, we demonstrate that the large $MV^{2+}$ can be reversibly intercalated in the PTCDA electrode with a reversible specific capacity of 105 mAh g$^{-1}$ and an excellent rate performance: 67 mAh g$^{-1}$ at the current rate of 2000 mA g$^{-1}$. Ex situ XRD and FTIR revealed the strong intercalation of $MV^{2+}$ with PTCDA. Some counter-intuitive results are obtained here. The use of large $MV^{2+}$ as the charge carrier does not compromise the capacity of PTCDA, and it does not decrease its rate capability either. The strong interaction between $MV^{2+}$, a molecular ion and a molecular structure of PTCDA raises the operation potential, instead of lowering it and mitigates the overpotential. Our results demonstrate that the ion/electrode intermolecular interactions have implications on the capacity, rate capability, and even cycle life stability. The first-principles DFT calculations suggest that $MV^{2+}$ ions are vertically inserted between the columns of the stacked PTCDA molecules with $MV^{2+}$ tilted with a relative angle of 45° with respect to the columns. The Bader charge analysis suggests that 0.7 electrons are transferred from PTCDA to each $MV^{2+}$ during the discharge, which suggests a good extent of polar covalent bonding between the guest and the host. This study provides some insights on a new direction to design battery chemistries by considering the ion/electrode non-ionic interactions. Furthermore, this study also suggests that electrochemical characterization, e.g., GCD, represent, in fact, powerful synthetic tools to assemble new supramolecular solids, which may have properties of values transcending different disciplines.

## Methods

**Materials**. The organic solid of 3,4,9,10-perylenetetracarboxylic dianhydride (PTCDA) and methyl viologen dichloride powder were purchased from Sigma Aldrich and investigated without further modification. To prepare the reduced $MV^{2+}$ ($MV^0$), excessive 1 M NaBH$_4$ (J. T. Baker) aqueous solution was added in 0.1 M MVCl$_2$ aqueous solution, and $MV^0$ powder was collected after drying the solution at 80 °C for one day.

**Materials characterization**. X-ray diffraction (XRD) patterns of the samples were collected on a Rigaku Miniflex Diffractometer with Cu Kα irradiation ($\lambda = 1.5406$ Å). FEI NOVA 230 field-emission scanning electron microscopy (FESEM) and FEI Titan 80–300 high-resolution transmission electron microscopy (TEM) were used to study the morphology and microstructure of the material. Thermogravimetric analysis (TGA, SDTQ600) was recorded from room temperature to 800 °C in air with a ramping rate of 10 °C min$^{-1}$. The content of carbon, hydrogen, and nitrogen in the electrodes were evaluated on a Mettler-Toledo CHN analyzer. Nitrogen adsorption-desorption isotherms were recorded at 77 K using a Micromeritics TriStar II 3020 instrument. Specific surface area was calculated through the Brunauer–Emmett–Teller (BET) method. Pore-size distribution curve was calculated from the isotherm by Barrett–Joyner–Halenda (BJH) algorithm. Fourier transform infrared spectroscopy (FTIR) was conducted on a NICOLET iS 10 FTIR spectrometer. The oxidation state of electrodes was probed by X-ray photoelectron spectroscopy (XPS) via a PHI Quantera Scanning ESCA Microprobe with a focused monochromatic Al X-ray source with a photon energy of 1486.6 eV.

The energy scale of the spectrometer is calibrated with Au $4f$ at 84.0 eV and Cu $2p_{3/2}$ at 932.7 eV.

**Electrochemical characterization**. The electrochemical performance was tested in three-electrode Swagelok cells (a T-cell), which comprised the PTCDA electrode as working electrode, activated carbon (AC) free-standing film as the counter electrode, Ag/AgCl electrode as the reference electrode, 0.1 M methyl viologen dichloride solution as the electrolyte, and Whatman filter paper as the separator. As for the control experiments, 0.1 M NaCl, 0.1 M KCl, 0.1 M MgCl$_2$, and 0.1 M NH$_4$Cl solution were chosen as the electrolytes. PTCDA was mixed with C-45 and polyvinylidene fluoride (PVdF) with a mass ratio of 7:2:1 in N-Methyl-2-Pyrrolidone (NMP) to make a slurry, which was then casted on carbon fiber paper as current collector and dried at 50 °C overnight. For ex situ measurements, titanium foil serves as current collector. To prepare the counter electrode, 70 wt.% AC, 20 wt.% C-45, and 10 wt.% polytetrafluoroethylene (PTFE) binder were mixed to make the self-standing films. Galvanostatic charge/discharge tests were carried out on a Maccor system at room temperature. Cyclic voltammetry (CV), rate performance, and galvanostatic intermittent titration technique (GITT) were carried out on an EC-Lab VMP3 instrument with a current pulse at 100 mA g$^{-1}$ for 2 min and a rest interval of 2 h. The ionic diffusion coefficient is determined by the following equation:

$$\tilde{D} = \frac{4}{\pi}\left(\frac{m_B V_M}{M_B S}\right)^2 \left(\frac{\Delta E_s}{\tau(dE_\tau/d\sqrt{\tau})}\right)^2 \left(\tau \ll \frac{L^2}{D}\right) \quad (1)$$

Where $V_M$, $M_B$ are the molar volume and the molecular weight of the materials, respectively. $m_B$ and $S$ represent the mass and active surface area of the electrode, respectively. $E_\tau$ represents the cell voltage during the current pulse for the time $\tau$. $\Delta E_s$ represents the change of the steady-state voltage of the cell over a single titration. $L$ is the average thickness of the electrode.

**Electrochemical quartz crystal microbalance measurement**. We conducted electrochemical quartz crystal microbalance (EQCM) measurement on a QCM 200 quartz crystal microbalance. To prepare working electrodes, 7 mg PTCDA, 2 mg C-45, and 4.5 mg PVdF were added in 0.5 mL NMP to form a slurry. After sonicating for 2 h, we sprayed the slurry onto a 1-inch quartz crystal disk (O100RX3, p/n 6–615 Ti/Au, 5 MHz). EQCM was recorded with cathodic CV scans. The mass change ($\Delta m$) of the electrode coated on the quartz crystal can be converted from the frequency change of the quartz resonator ($\Delta f$) by the Sauerbrey's equation:

$$\Delta m = \frac{\sqrt{\rho_q \mu_q}}{2f_0} * \Delta f = -C_f * \Delta f \quad (2)$$

where $\rho_q$, $\mu_q$, and $f_0$ are the density (2.648 g cm$^{-3}$), shear modulus (2.947 × 1011 g cm$^{-1}$ s$^{-2}$), and the fundamental resonance frequency of quartz, respectively. $\Delta m$ and $\Delta f$ are the mass change and frequency change, respectively. $C_f$ is the sensitivity factor, which is obtained by calculating the relation based on frequency and mass change (measured by balance) between the quartz crystal before and after coating. The value of the calibration constant used in this work is 14.6 ng/Hz, after collecting the data of frequency, and the molar weight of charge carrier (Mw) can be calculated by the following equation:

$$M_w = \frac{\Delta m\, n\, F}{\Delta Q} = \frac{C_f(-\Delta f)\, nF}{\Delta Q} \quad (3)$$

where $F$ is the Faraday constant (96485 C mol$^{-1}$), $n$ is the valence number of the ion, and $\Delta Q$ is the Coulombs of charges passed through during the CV processes.

**Theoretical calculations**. Density functional theory calculations (DFT) were performed using the Vienna ab initio simulation Package (VASP) with projector augmented wave (PAW) pseudopotentials and using the generalized gradient approximation (GGA) of Perdew-Burke-Ernzerhof (PBE) for the exchange-correlation function. Energy cutoff is 600 eV using a 2 × 2 × 1 Monkhorst-Pack reciprocal space grid of k-points for a single unit cell. Atomic coordinates were fully relaxed until all forces on each atom were below 0.02 eV Å$^{-1}$. The computational study performed here was to investigate the intercalation interaction between the electrochemically inserted MV$^{2+}$ and PTCDA crystals. XRD patterns were generated by Visualization for Electronic and Structure Analysis (VESTA) software package. To insert MV$^{2+}$ ions into the PTCDA without break/transformation of pristine structure, we applied the Avogadro software platform. For an expanded structure, we also used the same tool; however, we manually modified PTCDA crystals to make desired expanded structure.

## Data availability

Data that support the findings detailed in this study are available in the Supplementary Information and its Article or from the corresponding author upon reasonable request.

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

## Acknowledgements

X.J. is grateful to the U.S. National Science Foundation, Award Number 1551693 for the supports. F.D. thanks the support from the Joint Project between Jilin Province and Jilin University (SXGJQY2017-10), Science and Technology Development Project, Jilin Province (Grant Nos. 20180101211JC and 20190701020GH) and the Fundamental Research Funds for the Central University. J.L. gratefully acknowledges support from the U. S. Department of Energy (DOE), Office of Energy Efficiency and Renewable Energy, Vehicle Technologies Office. Argonne National Laboratory is operated for DOE Office of Science by UChicago Argonne, LLC, under contract number DE-AC02-06CH11357. Z. W. acknowledges the China Scholarship Council (CSC, No. 201706170130) for providing a scholarship for the exchange Ph.D. study at Oregon State University. We thank Professor Mas A. Subramanian and Professor Douglas A. Keszler for XRD measurements.

## Author contributions

X.J. conceived the concepts for the research project. X.J., F.D., P.A.G., J.L., and G.C. supervised the project. Z.W. and H.J. designed and performed electrochemical experiments and data analyses. W.S. conducted the computational simulation under the supervision from P.A.G.; X.W. participated in analyzing the experimental results. W.F.S. performed XPS tests. Z.W. drafted the manuscript.

## Additional information

**Competing interests:** The authors declare no competing interests.

