## [Peer Review File · Nature Communications]

Reviewers' comments:

Reviewer #1 (Remarks to the Author):

This manuscript reported the largest non-metal charge carrier, MV²⁺ for rechargeable batteries. MV²⁺ can be reversibly intercalated in the 3,4,9,10-perylenetetracarboxylic dianhydride (PTCDA) electrode with reversible high specific capacity and excellent rate performance. Meanwhile, ex situ XRD and FTIR to reveal the strong intercalation of MV²⁺ inside PTCDA. I think this is a nice work on the organic-aqueous electrolyte, though more data and evidence should be added. I would suggest to accept the manuscript after addressing the following issues.

1. In the traditional aqueous battery, the electrode host for various ions, including metal ions of Li⁺, Na⁺, K⁺, Mg²⁺, and non-metal hydronium and ammonium electrolyte display excellent properties in terms of specific capacity, structural stability, and so on. What is the advantage of the MV²⁺ compared with the other ions in terms of capacity, cycling stability, charge/discharge potential, or rate capability?
2. The Coulombic efficiency (CE) is not high in the initial cycles, but after five cycles the CE became higher. Please explain whether it's because the initial material is gradually disintegrating, the discharge capacity is higher than the charge capacity. How to improve it?
3. In the manuscript, the author gave two possible intercalated structures. Are there any other possible structures? How do the authors get these data? Besides comparing the energy of the unit cell, the charging analysis is also recommended to explain which structure is better.
4. In the manuscript, XRD results are the key evidence to determine the structure of the electrode intercalated with MV²⁺. In figure 2, it is quite obvious that the relative intensity ratio of (021) and (011) in experimental XRD is inconsistent with the simulated one of vertical case. The author explained this just by "the discrepancy in peak intensity may be caused by the lattice orientation of the tested thin electrode". Please give more solid and quantitative explanations about this to illustrate the reason why the relative intensity is quite different from the simulated one.
5. As is common to us, MV²⁺ ions are much larger than the commonly used metal ions such as Zn²⁺. However, the diffusivity is comparable to that of metal ions, which is quite counterintuitive. Please explain this further and give the mechanism behind.
6. The discussion on "reaction kinetics studies" is too general. These parts need to be re-written.
7. In page 4, the unit of the sentence "Thus, we estimate the capacity of reversible MV²⁺ storage as 78 mAh/g in the first cycle, which suggests that 0.57 MV²⁺ is reversibly incorporated per PTCDA." is inconsistent with another unit of specific capacity, for example, 137 mAh g⁻¹. Please check all the units used in the manuscript.
8. The form of authors' names in the 20th reference is not consistent with other references.

Reviewer #2 (Remarks to the Author):

This manuscript reports an interesting reversible intercalation of a new charge carrier, methyl viologen, into the aromatic PTCDA electrode, which exhibits a competitive electrochemical performance. Moreover, the fundamental mechanism studies show new insights of large molecular ions storage, involving the phase transition, the competition of intermolecular interaction. Therefore, I would recommend its acceptance after addressing the following points:

1. Line 21-22, the statement of 'This is the largest charge carries ever reported for batteries' is not appropriate. The co-intercalation chemistry in ether-based electrolyte may exhibit a larger charge carrier than methyl viologen (DOI: 10.1002/adfm.201402984).

2. For Figure 1a, on the one hand, the obvious plateaus at -0.5 V and -0.55 V can be observed during the first discharge while they disappeared during the second discharge; on the other hand, the charge plateaus keep the same during the first and subsequent cycles. Please comment on this point.
3. Generally, the coulombic efficiency of aqueous batteries should be very high. However, this work shows a poor coulombic efficiency, the authors need to discuss this point in the revised manuscript.

Reviewer #3 (Remarks to the Author):

This manuscript presents an in-depth study of the reversible topotactic (de)intercalation chemistry of a large size molecule (MV^{2+}) inside a highly crystalline PTCDA electrode. Electrochemical performance and the determination of charge carrier, interaction between MV^{2+} and MV^{2+} intercalated PTCDA, and reaction kinetics have been characterized and analyzed in detail. This work can extend the boundaries of redox topochemistry to large molecular ions and provide an inspiration of developing new energy storage devices. The results are interesting and solid. However, I cannot see any practical potential of this work in electrochemical energy storage. Many previous works have reported that conjugated carbonyl molecules such as PCDA can electrochemically store various small metal ions. Although larger organic ion can be stored in this work, what is the advantage of this system over previous ones. It is well known that larger ions will reduce the specific capacity normalize by the whole electrode and also lower the charge/discharge voltage. Meanwhile, how to construct the practical full battery based on this discovery. There are some other questions for the authors' attention.

(1) As PTCDA exhibits highly crystalline structure, its crystalline structure should be further characterized by high resolution TEM including the lattice size and selected area diffraction. In addition, the crystalline structure of PTCDA after the insertion of MV^{2+} should also be further characterized.

(2) Thermogravimetric analysis of the electrode after being fully discharged was performed to verify the intercalant, and the additional mass added during the discharge process was determined to be 15 wt.%, compared to the pristine PTCDA. However, nanomaterials are generally good adsorbents for some organic molecules. Therefore, the surface adsorption effect should be eliminated by suitable control experiment.

(3) The Coulombic efficiency of PTCDA electrode at 100 and 200 mA g⁻¹ is low, but it rises up to close to 100% when current densities increase, as shown in Fig. 1b and Fig. 4a. Authors should reasonably explain it.

(4) The cycling stability seems to be not good even tested in a short time. The author should demonstrate the long cycle life testing.

Response Letter to Reviewers' Comments

Response to Reviewer #1:

This manuscript reported the largest non-metal charge carrier, MV^{2+} for rechargeable batteries. MV^{2+} can be reversibly intercalated in the 3,4,9,10-perylenetetracarboxylic dianhydride (PTCDA) electrode with reversible high specific capacity and excellent rate performance. Meanwhile, ex situ XRD and FTIR to reveal the strong intercalation of MV^{2+} inside PTCDA. I think this is a nice work on the organic-aqueous electrolyte, though more data and evidence should be added. I would suggest to accept the manuscript after addressing the following issues.

Authors' Response: We thank the reviewer for the positive comments given and appreciate the revisions suggested, which are addressed below.

*1) In the traditional aqueous battery, the electrode host for various ions, including metal ions of Li^+ , Na^+ , K^+ , Mg^{2+} , and non-metal hydronium and ammonium electrolyte display excellent properties in terms of specific capacity, structural stability, and so on. What is **the advantage** of the MV^{2+} compared with the other ions in terms of capacity, cycling stability, charge/discharge potential, or rate capability?*

Authors' Response: We highly appreciate the reviewer for the insightful question. We have compared the electrochemical characteristics of PTCDA in the above-mentioned aqueous electrolytes, as detailed below:

To conduct the control experiments, we employed 0.1 M LiCl, 0.1 M NaCl, 0.1 M KCl, 0.1 M $MgCl_2$, 0.1 M HCl, and 0.1 M NH_4Cl solution as the electrolytes, respectively. Besides, note that the potential window in these systems was chosen to maximize the reversible capacity of PTCDA to perform a fair comparison, as shown in the following five figures.

First of all, in the aqueous electrolyte, lithium storage in PTCDA suffers from extremely large overpotentials, which triggers the irreversible hydrogen evolution reaction from on the second cycle (**Fig. a**). The humongous overpotentials may relate to the large ion size of the hydrated lithium ion in the aqueous electrolyte.

As for proton storage, with the diluted electrolyte of HCl, 0.1 M, the observed reversible capacity is exceedingly small, as shown in **Fig. b**.

Figure. Comparison of the GCD potential profiles for the storage of Li^+ and proton in PTCDA at a current rate of 100 mA g^{-1} .

Unlike Li^+ and proton, other cation charge carriers, including Na^+ , K^+ , Mg^{2+} , and NH_4^+ , can deliver comparable performance. The following details the comparison of their electrochemical properties with that of MV^{2+} .

As for capacity, MV^{2+} exhibits a comparable capacity in the much narrower voltage window (**Supplementary Fig. 10**).

As for charge/discharge potentials, the operation potential of MV^{2+} intercalation is higher than that of sodium-, potassium-, magnesium-, and ammonium-ion intercalation, where the capacity can be claimed at potentials well above that of the hydrogen evolution reaction.

As for polarization of the (de)intercalation reactions, that of MV^{2+} is also much smaller.

We have added the GCD potential profiles of PTCDA in different systems in **Supplementary Fig. 10** and added the highlighted comments to emphasize these points:

“Surprisingly, the reversible MV^{2+} intercalation exhibits a higher operation potential and a much smaller extent of polarization compared with several metal and non-metal ions such as sodium-, potassium-, magnesium- and ammonium-ions (Supplementary Fig. 10), suggesting superior thermodynamic and kinetic feasibility for PTCDA crystal to host MV^{2+} ions. The higher operation potential mitigates HER in acidic electrolytes.” [Page 6, Line 16-20]

Supplementary Figure 10. Comparison of the GCD potential profiles for the storage of Na^+ , K^+ , Mg^{2+} , NH_4^+ , and MV^{2+} in PTCDA at a current rate of 100 mA g^{-1} .

As for rate capability (Supplementary Fig. 16), when hosting MV^{2+} , the capacity retention of PTCDA is 67% when going up from a current rate of 100 mA g^{-1} to 2 A g^{-1} (Supplementary Fig. 17), which is higher than those of smaller metal-ion charge carriers including sodium-, potassium- and magnesium-ions. The unexpected results again indicate the good kinetic properties of MV^{2+} intercalation in PTCDA.

In contrast, ammonium-ion exhibits a better rate performance, which is understandable due to the smaller ionic radii. The comparison suggests an interesting phenomenon that the non-metal charge carriers may be good candidates for the high-power energy storage devices, while in-depth studies about the mechanism behind would be further investigated in future studies.

The figures have been added in the revised supplementary information, together with the highlighted discussion in the revised manuscript: “Interestingly, the rate performance of storing MV^{2+} in PTCDA is comparable with that of hosting metal-ions and ammonium-ions, as shown in Supplementary Fig. 16&17, which again indicates the good kinetics of MV^{2+} (de)intercalation.” [Page 13, Line 11-13]

Supplementary Figure 16. Rate capability tests for the storage of Na^+ , K^+ , Mg^{2+} , NH_4^+ and MV^{2+} in PTCDA at different current rates.

Supplementary Figure 17. The histogram illustrating the capacity retention of PTCDA in different battery systems during the current rates increasing from 100 mA g⁻¹ to 2 A g⁻¹.

As for cycling stability, PTCDA electrode demonstrates capacity retention of 86% after 100 cycles of MV²⁺ (de)intercalation at 200 mA g⁻¹, which is much more stable than Na⁺, K⁺, Mg²⁺ metal-ions, and ammonium-ion (**Supplementary Fig. 19**). To present the cycling stability of PTCDA in MV²⁺-battery system, we added the cycling tests in the revised manuscript, with the following discussion:

“Furthermore, the cycle stability of hosting MV²⁺ was also evaluated in comparison with the performance of other ions. As displayed in Supplementary Fig. 18, after 100 charge/discharge cycles at 200 mA g⁻¹, a charge capacity of 78 mA h g⁻¹ is retained, corresponding to capacity retention of 86%. Of note, the storage of MV²⁺ in PTCDA demonstrates much greater stability comparing with other ions, as illustrated in Supplementary Fig. 19.” [Page 13, Line 23-Page 14, Line 5]

Supplementary Figure 19. Cycling stability tests for the storage of Na⁺, K⁺, Mg²⁺, NH₄⁺ and MV²⁺ in PTCDA.

2) The Coulombic efficiency (CE) is not high in the initial cycles, but after five cycles the CE became higher. Please explain whether it's because the initial material is gradually disintegrating, the discharge capacity is higher than the charge capacity. How to improve it?

Authors' Response: We thank the reviewer for pointing this out. To monitor the structural change, we compared the XRD patterns of the PTCDA electrode after the first and fifth cycles. As shown in **Supplementary Fig. 13**, the XRD pattern of PTCDA after 5 cycles remains nearly the same as that of the pristine electrode, suggestive of the stable crystalline structure of the electrode.

The initially low CE may primarily relate to the trapping of MV^{2+} in the electrode structure. Notably, the increase of the CE mainly takes place from the first to the second cycle, which suggests the ion trapping being responsible for the initial low CE.

After the first five cycles, the discharge capacity is still higher than the charge capacity, giving rise to Coulombic efficiency still below 95%. This is attributed to HER on the PTCDA electrode.

We thank the reviewer for asking us on how to mitigate the HER. One approach is to raise the pH value of the electrolyte, which will push lower the onset potential of HER. Another approach may involve surface modification of the PTCDA particles by coating it with bismuth or lead nanoparticles.

We have added the XRD patterns in the revised supplementary information, as well as the corresponding explanations in the revised manuscript: “Moreover, the XRD peaks of the PTCDA electrode after 5 cycles are almost identical as those of the pristine electrode, as shown in Supplementary Fig. 13, suggesting the surprising structural stability of PTCDA crystal upon repetitive hosting MV^{2+} .” [Page 8, Line 10-13]

Supplementary Figure 13. *Ex situ* XRD patterns of the PTCDA electrode at OCV, the 1st cycle, and the 5th cycle.

3) In the manuscript, the author gave two possible intercalated structures. Are there any other possible structures? How do the authors get these data? Besides comparing the energy of the unit cell, the charging analysis is also recommended to explain which structure is better.

4) In the manuscript, XRD results are the key evidence to determine the structure of the electrode intercalated with MV^{2+} . In figure 2, it is quite obvious that the relative intensity ratio of (021) and (011) in experimental XRD is inconsistent with the simulated one of vertical case. The author explained this just by “the discrepancy in peak intensity may be caused by the lattice orientation of the tested thin electrode”. Please give more solid and quantitative explanations about this to illustrate the reason why the relative intensity is quite different from the simulated one.

Authors' Response to Questions 3&4: We highly appreciate the reviewer for the thoughtful suggestions. During our trials to find other possible structures, we found another structure that fits better with the experimental data. Hence, we revised the corresponding conclusion, as detailed below:

Since the MV^{2+} cation is much bigger than a typical metal ion, there are mainly two possible insertion sites: (1) Sandwiched between PTCDA molecules, (2) Normal to PTCDA planes between the columns of stacked PTCDA. Thus, the MV^{2+} molecules were placed in several plausible insertion sites, and then we let the systems to be fully relaxed to find an energetically favorable configuration, where we find out which structures match with the experimental XRD results better.

The pristine PTCDA structure was downloaded from the FindIt database, and the initial guesses of MV^{2+} -inserted structures were generated manually using an open source molecular builder, Avogadro. Firstly, we put MV^{2+} cations *in parallel* to the PTCDA molecules, followed by structural relaxation until the force on all atoms reduces to less than $0.02 \text{ eV}/\text{\AA}$. Fig. 4 displays geometries of sandwiched structures after energy minimization and the corresponding XRD patterns. Since the interplanar distance in pristine PTCDA is not big enough to host MV^{2+} ions, PTCDA molecules as well as MV^{2+} cations are forced to deform, as indicated in Fig. 4a, which is energetically unfavorable although the generated XRD peaks are in the proximity of the experimental one. To alleviate the disadvantage, we manually expanded the PTCDA interplanar space and relaxed the structure again (Fig. 4b). It still distorts the PTCDA domains, and the generated XRD patterns poorly match with the experimental one.

Figure 4. Simulated MV^{2+} inserted PTCDA geometries and corresponding XRD patterns after energy minimization: (a) parallel insertion into the pristine PTCDA domains; (b) parallel insertion after PTCDA interplanar distance extension along a -axis.

As for the second types of sites, we inserted MV^{2+} into the *crevices between the columns of the stacked PTCDA molecules*. Here, the angles between MV^{2+} and the PTCDA column vary from 0° (in original submission) to 45° to 90° (new calculations) to explore multiple insertion conditions, as shown in **Fig. 5**. All the MV^{2+} ions are aligned near the oxygen terminals of PTCDA molecules, owing to its hydrogen bonding with MV^{2+} . After structural relaxation, the generated XRD patterns have been compared to the experimental data.

As shown in **Fig. 6**, at the parallel-to-column position or at the normal-to-column position, the simulated XRD patterns fit the experimental results poorly. However, when we tilted MV^{2+} to be 45° , and surprisingly, the generated XRD pattern matches with the discharged-state XRD pattern very well (the stage 3 in **Figure 2**). Peaks split into (011), (02 $\bar{1}$) and (021), and they have similar relative intensity with the experimental output (**Fig. 6c**). There is some angle shift, which we postulate is due to the fact that the simulation is done ‘under vacuum conditions’, where the lattice d -spacings are different from the samples under ambient conditions. Generally, this tilted model can provide some valuable insights on how the MV^{2+} ions are stored and oriented with respect to PTCDA’s structure.

To validate this model, we extracted one MV^{2+} out of the unit cell to reach the stoichiometry of $3PTCDA:1MV^{2+}$ and simulated the corresponding XRD pattern (**Supplementary Fig. 14**). The simulated XRD peaks match well with stage 4 and stage 5 (the half-charged stage in **Figure 2**), which represents the de-insertion process. Therefore, MV^{2+} ions are most likely inserted at 45° into the crevices between the columns of the stacked PTCDA molecules.

Figure 5. Models of simulated MV^{2+} inserted PTCDA in vertical geometries from top view (top) and side view (bottom): (a) 0° ; (b) 90° ; (c) 45° . Angles are indicating relative slope to PTCDA domains.

Figure 6. Simulated XRD patterns of vertically inserted models: (a) 0° (b) 90° (c) 45° . Angles are indicating relative slope to PTCDA domains.

Supplementary Figure 14. The 45° vertical insertion model of partially filled PTCDA model (3 PTCDA: 1 MV²⁺) (a) The geometry after relaxation, (b) The corresponding XRD pattern.

In addition, we conducted *Bader charge analysis* on MV²⁺ to understand the interactions between the host and the inserted species, where in the 45° MV²⁺-inserted model, the oxidation state of MV is +1.31 instead of +2, where nearly 0.7 electrons are transferred from PTCDA to MV²⁺, and this may explain the driving force of vertical insertion.

Accordingly, we altered the *First-Principle Calculations* part, as highlighted in the revised manuscript. “In addition, we conducted Bader charge analysis on MV²⁺ to understand the interactions between the host and the inserted species, where in the 45° MV²⁺-inserted model, the oxidation state of MV is +1.31 instead of +2, where nearly 0.7 electrons are transferred from PTCDA to each MV²⁺, and this may explain the driving force of vertical insertion.” [Page 10-13]

5) As is common to us, MV²⁺ ions are much larger than the commonly used metal ions such as Zn²⁺. However, the diffusivity is comparable to that of metal ions, which is quite counterintuitive. Please explain this further and give the mechanism behind.

Authors' Response: We thank the reviewer for the thoughtful question. Generally, small multivalent metal ions in aqueous electrolytes, like Zn²⁺, are surrounded by water molecules to form bulky and clumsy solvation spheres, giving rise to large desolvation energy penalty at the electrode/electrolyte interfaces during ion insertion. On the other hand, for big ions with low charge density, like methyl viologen, the surrounding water molecules are largely hydrogen bonded, which are more mobile,³⁴ leading to a much lower dehydration energy for MV²⁺ before being inserted into the PTCDA crystals. This explains, albeit (de)intercalating a very large ionic charge carrier of MV²⁺, PTCDA still demonstrates comparable rate capability to that of metal ions, as shown in **Supplementary Fig. 16**,

To emphasize this point, we added the following statement in the revised manuscript, along with an additional reference: “Such an excellent diffusion behavior can be ascribed to the low dehydration energy penalty at the electrode/electrolyte interface. Specifically, in aqueous electrolytes, the water molecules that surround large ions, like MV²⁺, are largely hydrogen bonded and are more mobile than the case of small metal ions.³⁴ Hence, this facilitates the ion transport through the electrode/electrolyte interface.” [Page 13, Line 19-23]

34. Hribar, B., Southall, N.T., Vlachy, V. & Dill, K.A. How ions affect the structure of water. *J. Am. Chem. Soc.* **124**, 12302-12311 (2002)

6) The discussion on “reaction kinetics studies” is too general. These parts need to be re-written.

Authors' Response: We thank the reviewer for the kind suggestion. With additional data, discussion and explanations, we altered this part as follows:

Old: “We also investigated the rate capability of the MV²⁺ storage in PTCDA. As shown in Fig. 4a, at a high current rate of 2000 mA g⁻¹, a specific capacity of 66.7 mAh g⁻¹ can still be retained, and the shape of GCD potential profiles can be well preserved with the increasing

current rate (Fig. 4b). These features suggest a good diffusivity of MV^{2+} inside PTCDA despite MV^{2+} 's large size. Galvanostatic intermittent titration technique (GITT) measurements on the 31st charge process shows the overall trend of the diffusion coefficient (Fig. 4c, d). The diffusion coefficients are found to strongly depend on the specific SOC, and the minimal diffusivity values correspond to the potential plateaus of the charge process. Surprisingly, despite the large size of the MV^{2+} molecule, the diffusion coefficient is comparable with ions in aqueous ion batteries, such as Zn^{2+} as a charge carrier.^{32,33}

New: To further investigate the kinetic properties of storing MV^{2+} inside PTCDA crystals, we firstly tested the rate capability of the MV^{2+} -PTCDA electrode. As shown in Fig. 7a, at a high current rate of 2000 mA g^{-1} , a specific capacity of 66.7 mAh g^{-1} can still be retained, and the shape of the GCD potential profiles can be well preserved upon increasing the current rate (Fig. 7b). These features suggest fast transport of MV^{2+} in the cells despite MV^{2+} 's large size. Note that the disparity in the CE under different current rates can be attributed to the extents of the HER on the PTCDA electrode. As the current rate increases, the onset potential of HER is lowered due to polarization, thus inhibiting HER from taking place and raising the CE to nearly 100%. Interestingly, the rate performance of storing MV^{2+} in PTCDA is comparable with that of hosting metal-ions and ammonium-ions, as shown in Supplementary Fig. 16&17, which again indicates the good kinetics of MV^{2+} (de)intercalation. Galvanostatic intermittent titration technique (GITT) measurements were carried out on the 31st charge process to gain insight into the kinetics of MV^{2+} diffusion in PTCDA. Fig. 7c-d show the overall trend of the diffusion coefficient, which is found strongly dependent on the specific SOC, and the minimal diffusivity values correspond to the potential plateaus of the charge process. Surprisingly, despite the large size of the MV^{2+} molecule, the diffusion coefficient is comparable with ions in aqueous ion batteries, such as Zn^{2+} as a charge carrier.^{32,33} Such an excellent diffusion behavior can be ascribed to the low dehydration energy penalty at the electrode/electrolyte interface. Specifically, in aqueous electrolytes, the water molecules that surround large ions, like MV^{2+} , are largely hydrogen bonded and are more mobile than the case of small metal ions.³⁴ Hence, this facilitates the ion transport through the electrode/electrolyte interface. [Page 13-14]

7) In page 4, the unit of the sentence “Thus, we estimate the capacity of reversible MV^{2+} storage as 78 mAh/g in the first cycle, which suggests that 0.57 MV^{2+} is reversibly incorporated per PTCDA.” is inconsistent with another unit of specific capacity, for example, 137 mAh g^{-1} . Please check all the units used in the manuscript.

Authors' Response: The authors appreciate the reviewer for pointing this out. We went through the manuscript carefully and unified the unit of specific capacity into “ mAh g^{-1} ”. The change we made has been highlighted in the revised manuscript.

8) The form of authors' names in the 20th reference is not consistent with other references.

Authors' Response: We would like to thank the reviewer again for the careful reading and catching it up. We have corrected the form of authors' names, as highlighted in the revised manuscript.

Response to Reviewer #2:

This manuscript reports an interesting reversible intercalation of a new charge carrier, methyl viologen, into the aromatic PTCDA electrode, which exhibits a competitive electrochemical performance. Moreover, the fundamental mechanism studies show new insights of large molecular ions storage, involving the phase transition, the competition of intermolecular interaction. Therefore, I would recommend its acceptance after addressing the following points:

Authors' Response: We thank the reviewer for the positive comments given and appreciate the revision suggested, which are addressed below.

1) Line 21-22, the statement of ‘This is the largest charge carriers ever reported for batteries’ is not appropriate. The co-intercalation chemistry in ether-based electrolyte may exhibit a larger charge carrier than methyl viologen (DOI: 10.1002/adfm.201402984).

Authors' Response: We thank the reviewer for the insightful comment. We agree with the reviewer on this point. Indeed, it has been reported that in ether-based electrolytes, the solvent molecules can co-intercalate into graphite along with the Na-ions, forming ternary graphite intercalation compounds.^{R1-R3} Same behavior has also been demonstrated in the Li-system.^{R2,R4} Generally, the size of such complex intercalant is very large, while the specific radius varies with the characteristics of solvent molecules. On the other hand, in traditional aqueous system, the solvation sheath of bare ions is composed of small water molecules, resulting in relatively small hydrated ionic radii.^{R5}

Hence, we have changed the statement into “This is the largest **non-solvated intercalant** charge carrier ever reported for batteries”, as highlighted in the revised manuscript. [Page 2, Line 5]

In addition, to make the background more comprehensive, we added the above previous study as reference 8 [Kim, H., Hong, J., Park, Y.U., Kim, J., Hwang, I. & Kang, K. Sodium storage behavior in natural graphite using ether-based electrolyte systems. *Adv. Funct. Mater.* **25**, 534-541 (2015)]

R1. Kim, H., Hong, J., Park, Y.U., Kim, J., Hwang, I. & Kang, K. Sodium storage behavior in natural graphite using ether-based electrolyte systems. *Adv. Funct. Mater.* **25**, 534-541 (2015).

R2. Jache, B., Binder, J.O., Abe, T. & Adelhelm, P. A comparative study on the impact of different glymes and their derivatives as electrolyte solvents for graphite co-intercalation electrodes in lithium-ion and sodium-ion batteries. *Phys. Chem. Chem. Phys.* **18**, 14299-14316 (2016).

R3. Goktas, M. et al. Graphite as cointercalation electrode for sodium-ion batteries: electrode dynamics and the missing solid electrolyte interphase (SEI). *Adv. Energy Mater.* **8**, 1702724 (2018).

R4. Abe, T., Fukuda, H., Iriyama, Y. & Ogumi, Z. Solvated Li-ion transfer at interface between graphite and electrolyte. *J. Electrochem. Soc.* **151**, A1120-A1123 (2004).

R5. Nightingale Jr, E.R. Phenomenological theory of ion solvation. Effective radii of hydrated ions. *J. Phys. Chem.* **63**, 1381-1387 (1959).

2) For Figure 1a, on the one hand, the obvious plateaus at -0.5 V and -0.55 V can be observed during the first discharge while they disappeared during the second discharge; on the other hand, the charge plateaus keep the same during the first and subsequent cycles. Please comment on this point.

Authors' Response: We thank the reviewer for pointing this out. We attribute this phenomenon to the “conditioning” of the PTCDA crystalline electrode, where after the first cycle with higher overpotentials, the PTCDA’s structure and morphology are transformed to better accommodate the MV^{2+} molecules. As shown in **Supplementary Fig. 4**, after the initial charge & discharge cycle, the morphology of PTCDA turns from rod-like (shown in Supplementary Fig. 2) to flake-like, which is maintained through the following cycles. The decrease in the particle thickness can be beneficial for the intercalation of MV^{2+} ions. Note that such transformation is favorable since the average operation potential of MV^{2+} intercalation in the following cycles is increased, as also indicated by the smaller polarization.

We have added the *ex situ* SEM images in the revised supplementary information to emphasize this point, together with the corresponding discussion in the revised manuscript: “Note that the discharge profile in the first cycle shows higher polarization than in the following cycles. This may come from the significant “effort” of PTCDA crystals to accommodate MV^{2+} , as suggested by the significantly altered morphology after the 1st cycle (Supplementary Fig. 4). Importantly, such a conditioning process is desirable since the average operation potential of MV^{2+} intercalation in the following cycles is enhanced.” [Page 5, Line 3-8]

Supplementary Figure 4. *Ex situ* SEM images for the PTCDA electrode at different state of charge. (a, b) The 1st cycle; (c, d) The 5th cycle.

3) Generally, the coulombic efficiency of aqueous batteries should be very high. However, this work shows a poor coulombic efficiency, the authors need to discuss this point in the revised manuscript.

Authors' Response: We thank the reviewer for the insightful suggestion. Owing to the acidic nature of the electrolyte (pH = 3.5), although we chose a high cutoff potential to mitigate hydrogen evolution reaction (HER), HER still cannot be totally eliminated, especially at such a low current rate (100 mA g^{-1}). As a piece of supporting evidence, the coulombic efficiency of PTCDA in the diluted H_2SO_4 aq. at a pH of 3.5 is almost the same as that in MV^{2+} -based electrolyte (92.9%), as shown in **Supplementary Fig. 7**. On the other hand, if we increase the current rate, the coulombic efficiency can indeed rise to nearly 100% (**Fig. 7a**).

To address this concern, we added the following highlighted sentence to the revised manuscript: “Meanwhile, a similar CE (92.9%) in this case indicates that HER on PTCDA that can hardly be fully eliminated in this mildly acidic electrolyte, which is responsible for the low CE in MV^{2+} -based electrolyte.” [Page 5, Line 24-26]

Response to Reviewer #3:

This manuscript presents an in-depth study of the reversible topotactic (de)intercalation chemistry of a large size molecule (MV^{2+}) inside a highly crystalline PTCDA electrode. Electrochemical performance and the determination of charge carrier, interaction between MV^{2+} and MV^{2+} intercalated PTCDA, and reaction kinetics have been characterized and analyzed in detail. This work can extend the boundaries of redox topochemistry to large molecular ions and provide an inspiration of developing new energy storage devices. The results are interesting and solid. However, I cannot see any practical potential of this work in electrochemical energy storage. Many previous works have reported that conjugated carbonyl molecules such as PTCDA can electrochemically store various small metal ions. Although larger organic ion can be stored in this work, what is the advantage of this system over previous ones. It is well known that larger ions will reduce the specific capacity normalize by the whole electrode and also lower the charge/discharge voltage. Meanwhile, how to construct the practical full battery based on this discovery. There are some other questions for the authors' attention.

Authors' Response: We thank the reviewer for the encouraging comments and appreciate the revisions suggested. We have also addressed the concerns about the motivations of this work as well as the practical usefulness, as detailed below:

In the past few decades, tremendous effort has been devoted to the exploration of better *electrode materials* for rechargeable batteries. Despite the tremendous progress made, to focus only on electrode materials for one charge carrier of Li-ion has left narrow space for the battery performance to be further improved for metrics other than the energy density. Recently, new battery chemistries employing Na-, K-, Mg-, Al-, Zn-ion, or hydronium and ammonium as charge carriers emerge as new options for future energy storage. These progresses suggest that besides the investigation of *electrode materials*, the in-depth understanding of the interactions between electrode and charge carriers is also of great significance to meet the ever-increasing demand in energy storage.

Hence, in this presented contribution, we aim at exploring new designing principles of battery chemistry beyond purely ionic bonding that functions in metal-ion-based systems. In this study, MV^{2+} serves as a model molecular ion as a charge carrier. The goal is to see whether strong ion/electrode interaction with a large extent of covalent bonding could have implications in capacity, operation potential, and cycling. *The insights coming from this study, including the use of a large ion, e.g., MV^{2+} , as the charge carrier, does not compromise its capacity, contrary to the common sense, and does not compromise its rate capability either; the strong interaction between a molecular ion and a molecular electrode structure raises the operation potential, instead of lowering the charge/discharge voltage (Supplementary Fig. 10); there is a more preferred angle for a large molecular ion to insert into the electrode structure, which will provide guidance in future battery chemistry design.* We agree with the reviewer that this MV^{2+} /PTCDA may not be commercialized for practical applications; however, the new information from studying this ion/electrode pair will be highly valuable for the battery community.

We have added this section in the conclusions:

“Some counter-intuitive results are obtained here. The use of large MV^{2+} as the charge carrier does not compromise the capacity of PTCDA, and it does not decrease its rate capability

either. The strong interaction between a molecular ion and a molecular electrode structure raises the operation potential, instead of lowering it and mitigates the overpotential.”

“This study provides some insights on a new direction to design battery chemistries by considering the ion/electrode non-ionic interactions.” [Page 15, Line 2-5; Line 12-13]

The following is the comparison of MV^{2+} as a charge carrier with other metal ions, proton, and NH_4^+ . We employed 0.1 M LiCl, 0.1 M NaCl, 0.1 M KCl, 0.1 M $MgCl_2$, 0.1 M HCl, and 0.1 M NH_4Cl solution as the electrolytes, respectively. Besides, note that the potential window in these systems was chosen to maximize the reversible capacity of PTCDA to perform a fair comparison, as shown in the following five figures.

First of all, in the aqueous electrolyte, lithium storage in PTCDA suffers from extremely large overpotentials, which triggers the irreversible hydrogen evolution reaction from on the second cycle (**Fig. a**). The humongous overpotentials may relate to the large ion size of the hydrated lithium ion in the aqueous electrolyte.

As for proton storage, with the diluted electrolyte of HCl, 0.1 M, the observed reversible capacity is exceedingly small, as shown in **Fig. b**.

Figure. Comparison of the GCD potential profiles for the storage of Li^+ and proton in PTCDA at a current rate of $100 mA g^{-1}$.

Unlike Li^+ and proton, other cation charge carriers, including Na^+ , K^+ , Mg^{2+} , and NH_4^+ , can deliver comparable performance. The following details the comparison of their electrochemical properties with that of MV^{2+} .

As for capacity, MV^{2+} exhibits a comparable capacity in the much narrower voltage window (**Supplementary Fig. 10**).

As for charge/discharge potentials, the operation potential of MV^{2+} intercalation is higher than that of sodium-, potassium-, magnesium-, and ammonium-ion intercalation, where the capacity can be claimed at potentials well above that of the hydrogen evolution reaction.

As for polarization of the (de)intercalation reactions, that of MV^{2+} is also much smaller.

We have added the GCD potential profiles of PTCDA in different systems in **Supplementary Fig. 10** and added the highlighted comments to emphasize these points:

“Surprisingly, the reversible MV^{2+} intercalation exhibits a higher operation potential and a much smaller extent of polarization compared with several metal and non-metal ions such as sodium-, potassium-, magnesium- and ammonium-ions (Supplementary Fig. 10), suggesting superior thermodynamic and kinetic feasibility for PTCDA crystals to host MV^{2+} ions. The higher operation potential mitigates HER in acidic electrolytes.” [Page 6, Line 16-20]

Supplementary Figure 10. Comparison of the GCD potential profiles for the storage of Na^+ , K^+ , Mg^{2+} , NH_4^+ , and MV^{2+} in PTCDA at a current rate of 100 mA g^{-1} .

As for rate capability (**Supplementary Fig. 16**), when hosting MV^{2+} , the capacity retention of PTCDA is 67% when going up from a current rate of 100 mA g^{-1} to 2 A g^{-1} (**Supplementary Fig. 17**), which is higher than those of smaller metal-ion charge carriers including sodium-, potassium- and magnesium-ions. The unexpected results again indicate the good kinetic properties of MV^{2+} intercalation in PTCDA.

In contrast, ammonium-ion exhibits a better rate performance, which is understandable due to the smaller ionic radii. The comparison suggests an interesting phenomenon that the non-metal charge carriers may be good candidates for the high-power energy storage devices, while in-depth studies about the mechanism behind would be further investigated in future studies.

The figures have been added in the revised supplementary information, together with the highlighted discussion in the revised manuscript: “Interestingly, the rate performance of storing MV^{2+} in PTCDA is comparable with that of hosting metal-ions and ammonium-ions, as shown in Supplementary Fig. 16&17, which again indicates the good kinetics of MV^{2+} (de)intercalation.” [Page 13, Line 11-13]

Supplementary Figure 16. Rate capability for the storage of Na^+ , K^+ , Mg^{2+} , and NH_4^+ in PTCDA at different current rates.

Supplementary Figure 17. The histogram illustrating the capacity retention of PTCDA in different battery systems during the current rates increasing from 100 mA g^{-1} to 2 A g^{-1} .

As for cycling stability, PTCDA electrode demonstrates capacity retention of 86% after 100 cycles of MV^{2+} (de)intercalation at 200 mA g^{-1} , which is much more stable than Na^+ , K^+ , Mg^{2+} metal-ions, and ammonium-ion (**Supplementary Fig. 19**). To present the cycling stability of PTCDA in MV^{2+} -battery system, we added the cycling tests in the revised manuscript, with the following discussion:

“Furthermore, the cycle stability of hosting MV^{2+} was also evaluated in comparison with the performance of other ions. As displayed in Supplementary Fig. 18, after 100 charge/discharge cycles at 200 mA g^{-1} , a charge capacity of 78 mA h g^{-1} is retained, corresponding to capacity retention of 86%. Of note, the storage of MV^{2+} in PTCDA demonstrates much greater stability comparing with other ions, as illustrated in Supplementary Fig. 19.” [Page 13, Line 23-Page 14, Line 5]

Supplementary Figure 19. Cycling stabilities for the storage of Na^+ , K^+ , Mg^{2+} , NH_4^+ , and MV^{2+} in PTCDA.

To “*construct the practical full battery*”, we can benefit from the dual-ion battery (DIB) technology which is of wide interest in recent years. In DIBs, cations and anions are simultaneously (de)intercalate into/from host structure during the battery operation, generating high energy density. To date, several anion charge carriers have been successfully explored in non-aqueous system^[R1-R3] as well as aqueous system^[R4]. As for the anion host, graphite and porous carbon, for example, can serve as good cathodes for anion insertion/adsorption. Moreover, organic materials are also reported to be capable of hosting anions.^[R5-R7] Although in our case, the counter ion, Cl^- , may be challenging to insert into a cathode due to the large charge density, we believe that future studies attempting other anions may indeed realize the construction of full cells.

- R1. Ji, B., Zhang, F., Song, X. & Tang, Y. A novel potassium-ion-based dual-ion battery. *Adv. Mater.* **29**, 1700519 (2017).
- R2. Zhang, E. et al. A novel aluminum dual-ion battery. *Energy Storage Mater.* **11**, 91-99 (2018).
- R3. Zhang, M., Song, X., Ou, X. & Tang, Y. Rechargeable batteries based on anion intercalation graphite cathodes. *Energy Storage Mater.* **16**, 65-84 (2019).
- R4. Dong, X. et al. All-organic rechargeable battery with reversibility supported by “water-in-salt” electrolyte. *Chem. Eur. J.* **23**, 2560-2565 (2017).
- R5. Fan, L., Liu, Q., Xu, Z. & Lu, B. An organic cathode for potassium dual-ion full battery. *ACS Energy Lett.* **2**, 1614-1620 (2017).
- R6. Li, C. et al. Poly(N-vinylcarbazole) as an advanced organic cathode for potassium-ion-based dual-ion battery. *Electrochim. Acta* **297**, 850-855 (2019).
- R7. Dong, S. et al. A novel coronene/ $Na_2Ti_3O_7$ dual-ion battery. *Nano Energy* **40**, 233-239 (2017).

1) As PTCDA exhibits highly crystalline structure, its crystalline structure should be further characterized by high resolution TEM including the lattice size and selected area diffraction. In addition, the crystalline structure of PTCDA after the insertion of MV^{2+} should also be further characterized.

Authors' Response: The authors thank the reviewer for the insightful suggestion. We agree with the reviewer on this point that HRTEM is useful to understand the microstructure of the crystalline materials. However, generally, in the operation of HRTEM tests, the electron beam needs to penetrate the specimen for imaging (maximum 250 nm), while in our case, the particle size of PTCDA electrode is in micro-scale, thus the electrons could not go through the sample. Hence, instead of HRTEM, we have performed SAED studies to compare the crystalline structure of the MV^{2+} -intercalated PTCDA with the pristine PTCDA. As shown in **Supplementary Fig. 12**, in the discharged stage, the characteristic $(02\bar{1})$ peak of the new phase did appear in the pattern with a d -spacing of 9.2 Å, corroborating with the XRD pattern. The $(02\bar{1})$ plane is absent at the OCV and charged stages, which suggests the new phase formation. We have added the images to the revised supplementary information, as well as the following statements:

“Along this line, *ex situ* transmission electron microscopy (TEM) and selected area electron diffraction (SAED) studies are performed to gain insights into the microstructural transformation (Supplementary Fig. 12). At the end of discharge, the diffraction spot of the $(02\bar{1})$ peak emerges, which disappears during the following charge process. The results again confirm the reversible formation of the new phase.” [Page 8, Line 6-10]

Supplementary Fig. 12. *Ex situ* TEM images and SAED patterns of the PTCDA electrode at different state of charge: (a) OCV; (b) Discharged to -0.58 V; (c) Charged to 0.2 V.

2) Thermogravimetric analysis of the electrode after being fully discharged was performed to verify the intercalant, and the additional mass added during the discharge process was

determined to be 15 wt.%, compared to the pristine PTCDA. However, nanomaterials are generally good adsorbents for some organic molecules. Therefore, the surface adsorption effect should be eliminated by suitable control experiment.

Authors' Response: We appreciate the reviewer for pointing this out. As a control experiment, we dip the electrode in the MVCl₂ electrolyte for 3 days for adsorption, and performed another TGA test. As illustrated by the blue profile in the figure below, there is no additional weight loss of the immersed sample until 380 °C. The result suggests that MV²⁺ molecule would not adhere tightly on PTCDA surface. Accordingly, we altered **Fig. 1d** by the following figure in the revised manuscript.

Figure 1d. The TGA profiles of the pristine electrode (black), the immersed electrode (blue), and the discharged sample (olive) tested under air flow.

We added this section to the revised manuscript:

“As a control experiment, we immersed the PTCDA electrode in the MVCl₂ electrolyte for 3 days, and performed another TGA test afterwards. As illustrated by the blue profile in Fig. 1d, there is nearly no additional weight loss compared to the non-soaked electrode.” [Page 6, Line 10-13]

On the other hand, in general, the surface adsorption effect of a host material is proportional to its specific surface area. Hence, to evaluate the contribution of surface adsorption, the N₂ sorption isotherm was collected to determine the specific surface area and pore distribution of PTCDA (**Supplementary Fig. 9**). PTCDA contains a large number of macropores, which most likely represent the voids between PTCDA rods. Importantly, the Brunauer–Emmett–Teller (BET) surface area is only 13.29 m²/g, again indicating the minor contribution that surface adsorption effect makes here.

Supplementary Figure 9. (a) Nitrogen adsorption-desorption isotherms and (b) pore distributions of PTCDA.

To address this concern, we have also added the nitrogen adsorption-desorption isotherms in the revised supplementary information, together with the following highlighted discussion in the revised manuscript: “Besides, the Brunauer–Emmett–Teller (BET) surface area of PTCDA is 13.29 m²/g (Supplementary Fig. 9). Since the surface adsorption positively correlates with the specific surface area, it would not make a significant contribution in this case” [Page 6, Line 13-15]

3) The Coulombic efficiency of PTCDA electrode at 100 and 200 mA g⁻¹ is low, but it rises up to close to 100% when current densities increase, as shown in Fig. 1b and Fig. 4a. Authors should reasonably explain it.

Authors’ Response: We thank the reviewer for the insightful suggestion. The hydrogen evolution reaction (HER) on PTCDA is responsible for the low coulombic efficiency at low current rates. The control experiment of the PTCDA electrode in a diluted H₂SO₄ aq. with the same pH value can serve as a piece of evidence here. As shown in **Supplementary Fig. 7**, the coulombic efficiency of the PTCDA electrode in diluted H₂SO₄ electrolyte is similar with that in MV²⁺-based electrolyte.

As the current rate increases, the onset potential of HER is pushed lower due to the polarization of the reaction, thus inhibiting HER from taking place and raising the CE to nearly 100% (**Fig. 7a**).

To address this concern, we have revised the manuscript by adding the highlighted comments: “Note that the disparity in the CE under different current rates can be attributed to the extents of the HER on the PTCDA electrode. As the current rate increases, the onset potential of HER is lowered due to polarization, thus inhibiting HER from taking place and raising the CE to nearly 100%.” [Page 13, Line 8-11]

4) The cycling stability seems to be not good even tested in a short time. The author should demonstrate the long cycle life testing.

Authors' Response: We highly appreciate the reviewer for the kind suggestion. We have conducted longer cycling test on MV^{2+} (de)intercalation at a current rate of 200 mA g^{-1} . As displayed in **Supplementary Fig. 18**, PTCDA electrode demonstrates satisfactory cycle stability.

We have added the longer testing of cycle life in the revised supplementary information, with the highlighted comments in the revised manuscript:

“Furthermore, the cycle stability of hosting MV^{2+} was also evaluated in comparison with the performance of other ions. As displayed in Supplementary Fig. 18, after 100 charge/discharge cycles at 200 mA g^{-1} , a charge capacity of 78 mA h g^{-1} is retained, corresponding to capacity retention of 86%. Of note, the storage of MV^{2+} in PTCDA demonstrates much greater stability comparing with other ions, as illustrated in Supplementary Fig. 19.” [Page 13, Line 23-Page 14, Line 5]

Supplementary Figure 18. Cycle performance of PTCDA at 100 mA g^{-1} and 200 mA g^{-1} .

Finally, the authors hope to thank the editor for the efforts on this article.

Reviewers' comments:

Reviewer #1 (Remarks to the Author):

The revised manuscript now can be accepted for publication.

Reviewer #2 (Remarks to the Author):

The authors have addressed most of my comments. After reading the responses to the reviewers, there are some other questions the authors need to address.

1, As the unobvious changes of the relative intensity of FTIR peaks, it needs more evidences to further demonstrate the intercalation of MV2+.

2, On page 9, the authors speculated that the binding of MV2+ to PTCDA molecules may not be localized on the sites of C=O bonds; but being more delocalized, please illustrate it using the reaction equation in the form of the molecular structures.

3, For supplementary Figure 12 and 13, the peak intensity of the 5th discharged sample is significantly weaker than that of the first discharged sample while the charged states are similar. Whether it implies that the PTCDA molecular arrangement becomes increasingly disordered during the MV2+ intercalation process? If so, is it meaningful to study the intercalation angle of the MV2+? please comment on it. Additionally, the authors need to present XRD patterns and SAED of the 20th charged/discharged samples.

Reviewer #3 (Remarks to the Author):

This manuscript has been well revised and provided a detailed point-by-point response to the insightful comments of all three reviewers. However, I still have some concerns about this work.

1. Why the discharge curve shows no plateau while the charge curve shows obvious two plateaus?

2. As the maintext (results and discussion section) mentioned that the electrochemical test was performed in a electrolyte containing 3 M KCl and PTCDA can also store K+, the actual storage capacity of the MV2+ is probably be over-estimated.

3. The quantitative analysis of the content of MV2+ in the discharged state should be further studied by combustion analysis besides the TGA and discussed more detailedly.

4. Although the finding in this work is scientifically interesting, actually I can't see the potential application even the authors mentioned dual-ion batteries, since the PTCDA are usually used as cathode material and particularly the charge/discharge potential for MV2+ is relatively higher than other ion systems as the author claimed. So the fundamental understanding of the difference of this system between other ion systems is very important. In this aspect, why the MV2+ has higher charge/discharge potential and much better cycling stability than small ions. Why the rate capability of MV2+ is comparable/higher than small metal ions? The in-depth experimental and theoretical comparison analysis should be performed not just present some simple results and single analysis.

5. The electrochemical reaction in the system of PTCDA and MV2+ should be schematically depicted.

Response Letter to Reviewers' Comments

Response to Reviewer #1:

The revised manuscript now can be accepted for publication.

Authors' Response: We highly appreciate the reviewer for the positive comment on the revised manuscript.

Response to Reviewer #2:

The authors have addressed most of my comments. After reading the responses to the reviewers, there are some other questions the authors need to address:

Authors' Response: We thank the reviewer for the positive comments on the revised manuscript. We appreciate the revisions suggested, which are addressed below.

1) As the unobvious changes of the relative intensity of FTIR peaks, it needs more evidences to further demonstrate the intercalation of MV^{2+} .

Authors' Response: We thank the reviewer for catching the inconsistency of the relative intensity of the FTIR peaks. We apologize for the confusion. In the revised manuscript, we performed background correction on all the experimental results, as shown in the revised Fig. 3a. During discharge (MV^{2+} incorporation), the decrease in the peak intensity of C=O bonds suggests the reduction of the double bond.

Furthermore, we provide new evidence for the intercalation of MV^{2+} . We monitored the change of valence states of oxygen using *ex situ* XPS analysis (Fig. 3b, *newly added in this revision*). After the discharge process, the two peaks in O 1s spectrum, which can be assigned to C-O and C=O, shifted to lower binding energies, indicative of the reduction of the oxygen functional groups upon MV^{2+} insertion. After the following charge process, the peaks shifted back to the original state, again suggesting the reversible insertion of MV^{2+} .

We have updated the Fig. 3 in the revised manuscript, along with the following discussion added: "To better understand the electron transfer, we performed *ex situ* X-ray photoelectron spectroscopy (XPS) on O 1s to monitor the bonding evolution of oxygen upon discharge and charge processes, which turns out to be highly reversible (Fig. 3b). The spectra exhibit two peaks at 534 eV and 531.5 eV, denoting C-O and C=O, respectively. Both peaks red-shift in the discharged state, reflecting the increase in electron density on oxygen by accepting electrons." [Page 9, Line 8-13]

Moreover, quantitatively, we conducted CHN elemental analysis on the MV^{2+} -inserted PTCDA electrode. As shown in Supplementary Table 1, the content of nitrogen increases significantly in the discharged electrode and then decreases in the re-charged electrode. Since MV^{2+} molecule is the only nitrogen source in the complex, the result suggests the reversible intercalation of MV^{2+} once again. Furthermore, the molar ratio between the inserted MV^{2+} and PTCDA is calculated to be 0.579: 1, which is in well accordance with the GCD results.

We have added the CHN elemental analysis results and the calculation details in the revised Supplementary Information, as well as the discussion in the revised manuscript:

“Additionally, to further confirm the TGA results, CHN elemental analysis was also conducted. As shown in Supplementary Table 1, the results demonstrate that 0.579 MV²⁺ is incorporated per PTCDA molecule, which is in well accordance with the GCD results.”

[Page 6, Line 17-20]

Supplementary Table 1. CHN elemental analysis results of the pristine electrode, the discharged electrode and the re-charged electrode.

	C (%)	H (%)	N (%)
Pristine	74.49	2.411	0.55
Discharged	70.19	3.363	2.58
Charged	74.36	2.335	0.70

2) On page 9, the authors speculated that the binding of MV²⁺ to PTCDA molecules may not be localized on the sites of C=O bonds; but being more delocalized, please illustrate it using the reaction equation in the form of the molecular structures.

Authors' Response: We appreciate the reviewer for the kind suggestion. Based on the FTIR and XPS results, we added the schematic illustration of the possible reversible redox mechanism as Fig. 3c: “Combining the above discussion and the electrochemical results, the postulated redox mechanism is illuminated in Fig. 3c.” [Page 10, Line 10-11]

Figure 3. Analysis of the reaction mechanism during the electrochemical process. (a) FTIR spectra of the PTCDA electrode recorded at different SOC: OCV (black), the initial discharged (red), the charged state (blue) and FTIR spectra of methyl viologen dichloride powder (green) and its reduced form of MV⁰ (orange); (b) XPS O 1s spectrum of PTCDA

electrodes recorded at different SOC; (c) Possible reversible electrochemical redox mechanism of PTCDA electrode.

3) For supplementary Figure 12 and 13, the peak intensity of the 5th discharged sample is significantly weaker than that of the first discharged sample while the charged states are similar. Whether it implies that the PTCDA molecular arrangement becomes increasingly disordered during the MV^{2+} intercalation process? If so, is it meaningful to study the intercalation angle of the MV^{2+} ? please comment on it. Additionally, the authors need to present XRD patterns and SAED of the 20th charged/discharged samples.

Authors' Response: We appreciate the insightful question raised by the reviewer. We performed *ex situ* XRD test on the 20th discharged & charged samples, as shown in Supplementary Fig. 13. We found that the electrode became amorphized after the 20th discharge process and the peaks partially recovered after the 20th recharge, similar as that in the 5th cycle. Such phenomenon is often observed for the organic electrodes, which is possibly caused by the relatively weak interactions (such as *van der Waals* force in this case) among the organic molecules that comprise the crystalline structure,²⁸⁻³⁰ instead of the disintegration of the material.

Even so, the magnified XRD patterns (Supplementary Fig. 13b) of the discharged samples (after the 5th and 20th discharge) exhibit the same structural changes comparing to the initial discharged electrode except for the much weaker peak intensity. Especially, the $(02\bar{1})$ peak can still reversibly emerge upon the repeating MV^{2+} insertion. Hence, the intercalation model demonstrated by the DFT calculations is still tenable during the following cycles.

Moreover, we performed the SAED on the 20th discharged & recharged samples. However, due to the distraction of the brightness, the diffraction spots of those lattice planes with small *d*-spacings can hardly be revealed (See the figure below). In the given images, the dusky diffraction rings in the images could not provide valuable microstructural information of the crystals of the electrodes. We sincerely hope that the reviewer agrees with us that the *ex situ* XRD patterns can already address the concern.

Figure. *Ex situ* TEM images and SAED patterns of the PTCDA electrode in at different state of charge of 20th cycle: (a) Discharged to -0.58 V; (b) Charged to 0.2 V.

We have altered Supplementary Fig. 13 in the revised Supporting Information, together with the corresponding discussion: “Moreover, the evolution of the XRD peaks of the PTCDA electrode at the 5th cycle and the 20th cycle is almost identical as the first cycle except for the peak intensity, as shown in Supplementary Fig. 13, suggesting the surprising reversibility of structural changes of the PTCDA crystals upon repetitive hosting MV²⁺. It is worth noting that the discharged samples in the subsequent cycles all feature amorphization, which reflects the relatively weak interactions, i.e., *van der Waals* forces, among the PTCDA molecules that form the crystalline structure, as often observed for organic electrodes.²⁸⁻³⁰” [Page 8, Line 9-15]

28. Ma, T., Zhao, Q., Wang, J., Pan, Z. & Chen, J. A sulfur heterocyclic quinone cathode and a multifunctional binder for a high-performance rechargeable lithium-ion battery, *Angew. Chem. Int. Ed.* **55**, 6428-6432 (2016).
29. Hong, J., Lee, M., Lee, B., Seo, D. H., Park, C. B., & Kang, K. Biologically inspired pteridine redox centres for rechargeable batteries. *Nat. Commun.*, **5**, 5335 (2014).
30. Guo, C., Zhang, K., Zhao, Q., Pei, L., & Chen, J. High-performance sodium batteries with the 9, 10-anthraquinone/CMK-3 cathode and an ether-based electrolyte. *Chem. Commun.*, **51**, 10244-10247 (2015).

Supplementary Figure 13. (a) *Ex situ* XRD patterns of the PTCDA electrodes at OCV, the 1st cycle, the 5th cycle, and the 20th cycle; (b) Magnified *ex situ* XRD patterns of the PTCDA electrodes at discharged state in different cycles.

Response to Reviewer #3:

This manuscript has been well revised and provided a detailed point-by-point response to the insightful comments of all three reviewers. However, I still have some concerns about this work.

Authors' Response: The authors would like to thank the reviewer for the positive comments on the revised manuscript and appreciate the opportunity to clarify some issues of interests, which are presented below.

1) Why the discharge curve shows no plateau while the charge curve shows obvious two plateaus?

Authors' Response: We thank the reviewer for the insightful question. We attribute this phenomenon to the *asymmetric structural changes in crystallinity* during discharge and charge.

Firstly, in fact, as shown in Fig. 1a, the initial discharge potential profile demonstrates two obvious plateaus at -0.5 V and -0.55 V, where during this process, the morphology is significantly altered from rod-like to flake-like (Supplementary Fig. 4). During the second discharge, the plateaus merged into one slope-like plateau. Such transformation can be considered as a “conditioning process”, which is favorable since the average operation potential of MV^{2+} intercalation in the following cycles was increased as also indicated by the smaller polarization.

Secondly, why do the following charge processes in different cycles exhibit plateau profiles? The secret is revealed by the *ex situ* XRD patterns. We compared the XRD patterns of the electrodes at different SOC in the 1st cycle, the 5th cycle and the 20th cycle, as shown in Supplementary Fig. 13. For example, in the 5th cycle, the XRD pattern of the discharged electrode indicates a completely amorphous structure, which, however, amazingly returns back to a crystalline structure during the following charge process. Here are the intriguing nuances: the amorphous-to-crystalline structural change is featured as plateau profiles, while the crystalline-to-amorphous transition exhibits the characteristics of sloping profiles.

To address this concern, we added the following comments in the revised manuscript: “It is intriguing why the discharge profiles are slopes but the following charge processes depict plateau profiles. From the XRD patterns, as shown in Supplementary Fig. 13, the discharge process transforms the structure completely amorphous, which, however, returns back to a crystalline structure during the following charge process. The nuances are that the amorphous-to-crystalline structural change coincides with plateau profiles, whereas the crystalline-to-amorphous transition coexist with the sloping profiles.^{31,32} Such interesting structural changes reveal that the molecular structure of PCTDA is not only flexible to accommodate the incorporation of large ions but also elastic upon the removal of these ions.”

[Page 8, Line 16-23]

Supplementary Figure 13. (a) *Ex situ* XRD patterns of the PTCDA electrodes at OCV, the 1st cycle, the 5th cycle and the 20th cycle; (b) Magnified *ex situ* XRD patterns of the PTCDA electrodes at discharged state in different cycles.

31. Zhao, H. et al. Organic thiocarboxylate electrodes for a room-temperature sodium-ion battery delivering an ultrahigh capacity. *Angew. Chem. Int. Ed.* **56**, 15334-15338 (2017).
32. Lee, H. H., Park, Y., Shin, K.-H., Lee, K. T., & Hong, S. Y. Abnormal excess capacity of conjugated dicarboxylates in lithium-ion batteries. *ACS Appl. Mater. Interfaces* **6**, 19118-19126 (2014).

2) As the main text (results and discussion section) mentioned that the electrochemical test was performed in an electrolyte containing 3 M KCl and PTCDA can also store K^+ , the actual storage capacity of the MV^{2+} is probably be over-estimated.

Authors' Response: We thank the reviewer for pointing this out. We understand that the reviewer is concerned with the 3 M KCl in the reference electrode. To address this concern, we replaced the supporting 3 M KCl solution with 0.1 M $MVCl_2$ solution in the reference electrode, and performed another GCD tests as the control experiment. Note that since the 0.1 M supporting solution changes the standard of potential of Ag/AgCl to 0.28 V vs. SHE, we adjusted the voltage window for the GCD tests to -0.65 V~0.2 V accordingly. As shown in the following figure, PTCDA can deliver a charge capacity of 93 mAh g^{-1} , similar with our previous results.

Figure. GCD potential profiles of PTCDA in the initial three cycles, using 0.1 M MVCl₂ as the supporting solution in the reference electrolyte.

3) *The quantitative analysis of the content of MV²⁺ in the discharged state should be further studied by combustion analysis besides the TGA and discussed more detailedly.*

Authors' Response: We thank the reviewer for the insightful suggestion. We conducted CHN elemental analysis on the discharged sample, and the results are listed in Supplementary Table S1. Clearly, after the discharge process, the content of nitrogen rises significantly, indicative of the insertion of MV²⁺ molecules in the PTCDA crystals. According to the mass ratio between carbon and nitrogen in the discharged electrode sample, the molar ratio between the inserted MV²⁺ and PTCDA is calculated to be 1: 0.579, which is consistent with the ratio estimated by the TGA results (1: 0.53) as well as the GCD results (1: 0.57).

We have added the CHN element analysis results and the calculation detail in the revised Supplementary Information, together with the discussion in the revised manuscript: “Additionally, to further confirm the TGA results, CHN elemental analysis was also conducted. As shown in Supplementary Table 1, the results demonstrate that 0.579 MV²⁺ is inserted per PTCDA molecule, which is in well accordance with the GCD results.” [Page 6, Line 17-20]

Supplementary Table 1. CHN elemental analysis results of the pristine electrode, the discharged electrode and the re-charged electrode.

	C (%)	H (%)	N (%)
Pristine	74.49	2.411	0.55
Discharged	70.19	3.363	2.58
Charged	74.36	2.335	0.70

4) Although the finding in this work is scientifically interesting, actually I can't see the potential application even the authors mentioned dual-ion batteries, since the PTCDA are usually used as cathode material and particularly the charge/discharge potential for MV^{2+} is relatively higher than other ion systems as the author claimed. So, the fundamental understanding of the difference of this system between other ion systems is very important. In this aspect, why the MV^{2+} has higher charge/discharge potential and much better cycling stability than small ions. Why the rate capability of MV^{2+} is comparable/higher than small metal ions? The in-depth experimental and theoretical comparison analysis should be performed not just present some simple results and single analysis.

Authors' Response: We highly appreciate the thoughtful questions raised by the reviewer.

About the charge/discharge potential, we have added the following section: “A question is why the discharge potential for PTCDA to host MV^{2+} is higher than hosting other metal ions. There are two major factors that determine the potential of ion insertion: (1) the desolvation energy of the ion charge carrier from the aqueous electrolyte and (2) the binding energy between the inserted ions and the host structure. The desolvation energy of the much larger MV^{2+} is understandably lower than the smaller metal ions, where by compensating a smaller energy penalty for desolvation, the insertion potential is raised. On the other hand, MV^{2+} is redox active, where based on Bader charge analysis on the inserted MV^{2+} (45° tilted), the oxidation state of MV is +1.31 instead of +2, where nearly 0.7 electrons are transferred from PTCDA to one MV^{2+} . Such an electron transfer from the host to the inserted ion suggests a high extent of the covalent bonding between the MV charge carrier and the PTCDA host. The strong guest-host bonding would lower the total energy of the electrode system, thus raising the insertion potential. Therefore, these two factors may explain why the discharge potential of MV^{2+} is higher than the insertion of smaller metal ions.” [Page 16, Line 1-14]

About the cycling stability and rate capability, we have the following section: “Another interesting aspect is that the storage of much larger MV^{2+} in PTCDA exhibits more stable cycling performance and slightly better rate capability compared to other smaller metal ions (Supplementary Fig. 16&17). This phenomenon points to the solvated ion size, which we postulate is responsible for our observation. Generally, due to the high charge density, small metal ions are surrounded by water molecules to form bulky and clumsy solvation spheres, giving rise to large desolvation energy penalty at the electrode/electrolyte interfaces during ion insertion. For instance, the smallest metal ion, lithium ion, suffers from large overpotentials in the aqueous electrolytes due to the large size of the hydrated lithium ion, which cannot even survive the second cycle of the (de)insertion into the PTCDA electrode (see Supplementary Fig. 20). On the other hand, MV^{2+} are vertically inserted into the slits of the herringbone structure of PTCDA, where the effective ionic size is essentially the thickness of the planar molecule, being much smaller than the hydrated spheres of metal ions. This may facilitate the ionic transportation in the host crystals, and thus promote the better rate capability and cycling stability.” [Page 16, Line 15-27]

Supplementary Figure 20. GCD potential profiles for the storage of Li^+ in PTCDA at a current rate of 100 mA g^{-1} .

5) The electrochemical reaction in the system of PTCDA and MV^{2+} should be schematically depicted.

Authors' Response: We thank the reviewer for the kind suggestion. Based on the FTIR, XPS results and electrochemical tests, we illustrated the electrochemical reaction as Fig. 3c in the revised manuscript.

Finally, the authors would like to thank the editor and the reviewers for the time and effort on this article.

Reviewers' comments:

Reviewer #2 (Remarks to the Author):

The authors have well addressed all the issues, i would recommend its acceptance.

Reviewer #3 (Remarks to the Author):

This manuscript provides a further revision and response to the comments of the reviewers, which make this work stronger. However, the response to the fundamental understanding of the difference between MV^{2+} and other ions is still a little weak. The authors are suggested to use theoretical methods to analyze the electrochemical intercalation process involving the desolvation. Another question is how the charge transfer between negative PTCDA and MV^{2+} form? If the PTCDA was reduced first and then give electron to the intercalated MV^{2+} , is it possible the MV^{2+} can be reduced directly? The claimed covalent bonding between PTCDA and MV^{2+} seems to be a little arbitrary. In addition, it will be more interesting and important to test larger molecular cation to explore the upper limit of this intercalation reaction.

Response Letter to Reviewers' Comments

Response to Reviewer #2:

The authors have well addressed all the issues, I would recommend its acceptance.

Authors' Response: We highly appreciate the reviewer for the positive comment on the revised manuscript.

Response to Reviewer #3:

This manuscript provides a further revision and response to the comments of the reviewers, which make this work stronger. However, the response to the fundamental understanding of the difference between MV^{2+} and other ions is still a little weak. The authors are suggested to use theoretical methods to analyze the electrochemical intercalation process involving the desolvation. Another question is how the charge transfer between negative PTCDA and MV^{2+} form? If the PTCDA was reduced first and then give electron to the intercalated MV^{2+} , is it possible the MV^{2+} can be reduced directly? The claimed covalent bonding between PTCDA and MV^{2+} seems to be a little arbitrary. In addition, it will be more interesting and important to test larger molecular cation to explore the upper limit of this intercalation reaction:

Authors' Response: We appreciate the reviewer for the positive comments on the revised manuscript. We are grateful for the further thought-provocative comments from the reviewer, which continue to help improve the quality of this work. We have revised the manuscript accordingly, as explained in the following.

1). The authors are suggested to use theoretical methods to analyze the electrochemical intercalation process involving the desolvation.

Authors' Response: We thank the reviewer for this thoughtful suggestion. We appreciate the views of the reviewer about desolvation. We hope the reviewer could kindly consider that the theoretical study of the desolvation of MV^{2+} in the aqueous system is extremely hard to achieve. Our diluted solution (0.1 M) needs a great amount of water molecules plus the $MVCl_2$ molecules, which means that approximately more than 500 atoms are required in a simulation box to observe the reasonable results, which necessitates tremendous computational resources and time. Such computation of 500 atoms itself is unprecedented in the field of batteries, besides, the mobile water molecules in the system may destroy the periodicity of the atomic arrangement to a great extent, which makes the computation even harder. Therefore, the suggested theoretical work by itself is a full paper just for MV^{2+} , which is beyond the scope of this work.

We are grateful for the insightful suggestion. It is definitely a great topic to study separately to understand how the big molecular cations behave differently from the metal ions in the aqueous system, and the impact of desolvation structures on electrochemical properties for future studies.

To clarify this point, we have revised this section.

“There are two major factors that determine the potential of ion insertion: (1) the desolvation energy of the ion charge carrier from the aqueous electrolyte and (2) the binding energy

between the inserted ions and the host structure. The desolvation energy of the much larger MV^{2+} is understandably lower than the smaller metal ions, where by compensating a smaller energy penalty for desolvation, the insertion potential is raised.”

to the following:

There are two major factors that determine the potential of ion insertion: (1) the desolvation energy of the ion charge carrier from the aqueous electrolyte and (2) the binding energy between the inserted ions and the host structure. The desolvation energy of the much larger MV^{2+} is understandably lower than the smaller metal ions, where by compensating a smaller energy penalty for desolvation, the insertion potential is raised. Note that it can be very interesting to theoretically understand how the desolvation of large molecular ions such as MV^{2+} affects the electrochemical properties of host electrodes in comparison with smaller metal ions. However, considering the scope of this work and the primary theme about what takes place after MV^{2+} ions migrate into the PTCDA structures, such theoretical studies can be conducted in the future. (Page 16, Line 7-Page 17, Line 3)

2). Another question is how the charge transfer between negative PTCDA and MV^{2+} form? If the PTCDA was reduced first and then give electron to the intercalated MV^{2+} , is it possible the MV^{2+} can be reduced directly? The claimed covalent bonding between PTCDA and MV^{2+} seems to be a little arbitrary.

Authors' Response: We thank the reviewer for raising this thoughtful point. In a nutshell, it is not likely that MV^{2+} is reduced to MV^+ before being inserted into PTCDA. This conclusion is deduced from the computational results.

First of all, it is worth noting that, in the computation environment, we put all molecules at once and set the total number of electrons in the system, *i.e.*, for the state of charge of being fully discharged. For any computation, the overall charge of the system is neutral.

We performed the Bader charge analysis on the cation for the fully discharged state of charge. If there were no charge transfer from PTCDA to MV^{2+} , the MV^{2+} is supposed to have +2 charges, but it turns out to have +1.31 charges by the Bader charge analysis, which means 0.69 electrons are transferred from PTCDA to MV^{2+} .

In this revision, we did another Bader charge analysis on the PTCDA host, which is supposed to have -1 charge according to the ratio of 1/2 between MV^{2+} and PTCDA, but the Bader charge of PTCDA turns out to be -0.69. The result suggests that 0.62 electrons in total are transferred to MV^{2+} , corroborating the previous calculation.

It seems that MV^{2+} should be reduced first to $MV^{1.31+}$, and then PTCDA is reduced afterward from the above Bader charge calculations. However, as a single discrete molecule of MV^{2+} , it cannot possibly take in $0.69e^-$ from the surface of PTCDA electrode because electrons are indivisible in electrochemical reactions.

Therefore, it is more likely that during the MV^{2+} insertion, electrons are received by PTCDA molecules first, and the PTCDA molecules would pass these electrons to the intercalated MV^{2+} via the charge transfer, instead of the direct reduction of MV^{2+} itself on the surface of PTCDA.

We have added the following section in the revised manuscript:

“In addition, we conducted Bader charge analysis on MV^{2+} to understand the interactions between the host and the inserted species, where in the 45° MV^{2+} -inserted model, the oxidation state of MV is +1.31 instead of +2, where nearly 0.7 electrons are transferred from PTCDA to one MV^{2+} , and this may explain the driving force of vertical insertion. We also did the Bader charge analysis on the PTCDA host, which would be -1 according to the ratio of 1/2 between MV^{2+} and PTCDA if there were no charge transfer. The calculated Bader charge of PTCDA turns out to be -0.69. Thus, 0.31 electrons are transferred from each of the two PTCDA molecules to MV^{2+} and the Bader charge of MV^{2+} is 1.38, corroborating the above calculation. It seems that MV^{2+} should be reduced first to $MV^{1.31+}$ or $MV^{1.38+}$, and then PTCDA is reduced afterward from the above Bader charge calculations. However, as a single discrete molecule of MV^{2+} , it cannot possibly take in $0.69e^-$ from the surface of PTCDA electrode because electrons are indivisible in electrochemical reactions. Therefore, it is more likely that during the MV^{2+} insertion, electrons are received by PTCDA molecules first, and the PTCDA molecules would pass these electrons to the intercalated MV^{2+} via the charge transfer, instead of the direct reduction of MV^{2+} itself on the surface of PTCDA.” (Page 13, Line 11-22)

As for the claimed covalent bonding, we have changed to “polar covalent bonding”, which can better describe the situation.

3). In addition, it will be more interesting and important to test larger molecular cation to explore the upper limit of this intercalation reaction.

Authors' Response: We thank the reviewer for raising this wonderful suggestion. We tried a larger molecule dication, ethyl viologen, as the charge carrier in this system. As shown in the following Supplementary Figure 21 (new in this revision), the ethyl viologen molecule can also be reversibly inserted into the PTCDA crystals with the similar reversible capacity.

Supplementary Figure 21. GCD potential profiles for the storage of ethyl viologen in PTCDA at a current rate of 100 mA g^{-1} .

We have added the following sections to address this comment.

“As the last note, an intriguing question is whether molecules larger than MV^{2+} can be employed as the ion charge carriers for the PTCDA electrode. To shed some light on this interesting question, we tested the electrochemical (de)insertion of ethyl viologen (EV^{2+}) into the same PTCDA electrode. Remarkably, the reversible capacity is around 125 mAh g^{-1} , which is even slightly higher than MV^{2+} (Supplementary Fig. 21). Thus, we stay optimistic that even larger molecules may be inserted into PTCDA, where such an electrochemical reaction is, in fact, a powerful synthesis method of assemblies of supramolecular solids.” (Page 17, Line 24-30).

In the Conclusions section: “Furthermore, this study also suggests that electrochemical characterization, *e.g.*, GCD, represent, in fact, powerful synthetic tools to assemble new supramolecular solids, which may have properties of values transcending different disciplines.” (Page 18, Line 15-18)

Finally, we really hope to thank the Review 3 for raising excellent questions to help this manuscript better.

We sincerely thank Editor Zhang and the editorial team for the kind support.

REVIEWERS' COMMENTS:

Reviewer #3 (Remarks to the Author):

This manuscript has been further revised with a detailed response. This work can be accepted for publication. It is very interesting that the PTCDA can store larger organic molecule, thus I suggest the authors to carry out deep investigation in their subsequent research. In addition, the claim of MV²⁺ as the largest/longest molecule insertino charge carrier throughout the manuscript should be examined.

Response Letter to Reviewers' Comments

Response to Reviewer #3:

This manuscript has been further revised with a detailed response. This work can be accepted for publication. It is very interesting that the PTCDA can store larger organic molecule, thus I suggest the authors to carry out deep investigation in their subsequent research. In addition, the claim of MV^{2+} as the largest/longest molecule insertion charge carrier throughout the manuscript should be examined.

Authors' Response: We highly appreciate the reviewer for the positive comment on our revised manuscript. We would also like to thank the reviewer for the excellent suggestion on the follow-up work. Furthermore, we have gone through the manuscript carefully and unified the description of this molecule into “the largest”:

Page 2, Line 4: longest — largest